# Multipolar electric and magnetic contributions to sum-frequency generation spectra reveal biaxial interfacial water structure

Louis Lehmann [1], Maximilian R. Becker [1], Lucas Tepper [1],
Alexander P. Fellows [2], Álvaro Diaz Duque[2], Martin Thämer [2], Martin Wolf [2] &
Roland R. Netz [1]

Liquid interfaces play central roles in biological and physicochemical processes. Sum-frequency-generation (SFG) spectroscopy is intrinsically interface-specific, but the insight gained from SFG spectra into molecular interfacial structure has been limited since spectral analysis is usually done within the electric-dipole approximation, neglecting higher-order multipole contributions. Here we introduce a general framework that includes electric and magnetic multipoles for calculating SFG spectra from molecular simulations, achieving quantitative agreement between predicted and experimental SFG spectra of the air-water interface. We show that the electric-dipole approximation breaks completely down in the water bending region and remains only qualitatively valid in the OH-stretch region. When accounting for multipole contributions, the analysis of the SFG bending band reveals pronounced biaxial water ordering in a triple-layer structure with a width of only about 0.8 nanometers. By resolving a fundamental limitation of the interpretation of SFG spectroscopy, our framework allows for the detailed extraction of interfacial molecular ordering from SFG spectra.

Interfaces are crucial in biological[1–5] and physicochemical[6–10] applications. In particular, the air–water interface plays an essential role in catalysis[11–15], atmospheric[16–18], and prebiotic[19,20] chemistry. Undoubtedly, understanding the microscopic structure of interfaces is essential. The inhomogeneous interfacial region is only a few angstroms thick[21–24], or up to a few nanometers if the interface is charged and the salt concentration is small[25,26], and thus much thinner than the bulk region. This presents significant challenges in spectroscopy, as signals from the interface must be separated from the dominating bulk signals.

An elegant solution is to measure the second-order susceptibility, which is typically assumed to be non-zero only when the spatial inversion symmetry is broken, using sum-frequency-generation (SFG) spectroscopy. Normally, the bulk medium is isotropic and, therefore, the interface's structure accounts for a significant part of the SFG spectrum. The interpretation of experimental spectra usually assumes that second-order radiation originates from an interfacial electric dipole layer oscillating at the sum of the applied beams' frequencies. In this approximation, the SFG signal is determined by a second-order electric dipole susceptibility[27,28] and model calculations allow for the determination of the orientation of chemical bonds[29–31], the effective interfacial dielectric constant[32–34], the interfacial thickness[24,35], the surface potential[36–39], and water structure around macromolecules[40,41] from experiments.

[1]Department of Physics, Freie Universität Berlin, Berlin, Germany. [2]Fritz-Haber-Institut der Max-Planck-Gesellschaft, Berlin, Germany.
e-mail: rnetz@physik.fu-berlin.de

However, this dipole-layer picture neglects higher-order multipole contributions present in the SFG signal. Those multipole contributions to SFG spectra are proportional to higher-order response functions that are non-zero even in isotropic bulk media[42–46] and cannot be easily experimentally separated from the measured signal. To determine the structure of an interface from SFG spectra, one needs to identify the electric dipole component, which serves as a fingerprint of the interface structure. Thus, accurate theoretical predictions of higher-order multipole contributions are essential.

Off-resonant multipole contributions, measured in second-harmonic generation, were predicted in previous works[47,48]. Resonant multipole SFG contributions were estimated using normal-mode calculations[49] and found to be significant for the water bending band[50]. However, normal-mode methods involve rather drastic approximations, including locality assumptions, and cannot predict the spectral line shape. Experimentally, the multipolar origin of the water bending band has been investigated with SFG at surfactant monolayers[51–53] and combined SFG and difference-frequency-generation (DFG) measurements[54], yielding conflicting results. Many studies attribute the bending band to the electric dipole component of the SFG signal[51,55–59], in contrast to our findings.

We introduce a method, based on time-dependent perturbation theory[60,61], that enables the quantitative prediction of multipolar SFG spectral contributions and thereby constitutes a crucial step toward an accurate determination of the interfacial structure. We compute all relevant multipole contributions, including the magnetic dipole contribution; we find the latter to be sizable and of similar intensity as the electric dipole contribution in the bending band. Following the work of Guyot-Sionnest and Shen[62], we formulate the multipolar second-order electric current density as a response to electric displacement ($D$) fields perpendicular to the interface and electric ($E$) fields parallel to the interface, both of which are spatially constant on the relevant length scale. In doing so, we eliminate the need to make ad-hoc assumptions on the interface structure and observe quantitative agreement between simulated and experimental SFG spectra, with significant multipole contributions. Including multipole contribution in the analysis reveals that interfacial water exhibits an approximately 8 Å thick biaxial triple-layer structure.

## Results

### Quantitative comparison of SFG spectra with experiments

In Fig. 1, we present SFG spectra $\widetilde{S}^{(2)}_{ijk}$ for the polarization combinations $yyz$ and $zzz$, as defined in Eqs. (8) and (14). Here, the indices $ijk$ denote the polarizations of the second-order electric current density, which produces the observed radiation, the VIS field, and the IR field, respectively. The theoretical framework for predicting multipolar SFG spectra is described in detail in Supplementary Information (SI) Sections I–VI.

We observe quantitative agreement between our predictions (red lines) and experiments (black lines) in the bending and stretching bands in Fig. 1c, g, h, k. We analyze the SFG spectrum by dissecting it into its multipole contributions $\widetilde{S}^{(2)}_{ijk} = \widetilde{S}^{(2,\mathrm{DD})}_{ijk} + \widetilde{S}^{(2,\mathrm{DQ})}_{ijk} + \widetilde{S}^{(2,\mathrm{Q})}_{ijk} + \widetilde{S}^{(2,\mathrm{M})}_{ijk}$. This decomposition is described in the section "Methods" "Molecular multipole contributions." Here, $\widetilde{S}^{(2,\mathrm{DD})}_{ijk}$ and $\widetilde{S}^{(2,\mathrm{DQ})}_{ijk}$ are the pure electric dipole contribution and the electric dipole - electric quadrupole cross contribution, respectively. Importantly $\widetilde{S}^{(2,\mathrm{DD})}_{ijk}$ describes the interfacial structure, while $\widetilde{S}^{(2,\mathrm{DQ})}_{ijk}$ is created by the linear response of electric dipoles to fields from second-order electric quadrupoles; $\widetilde{S}^{(2,\mathrm{Q})}_{ijk}$ and

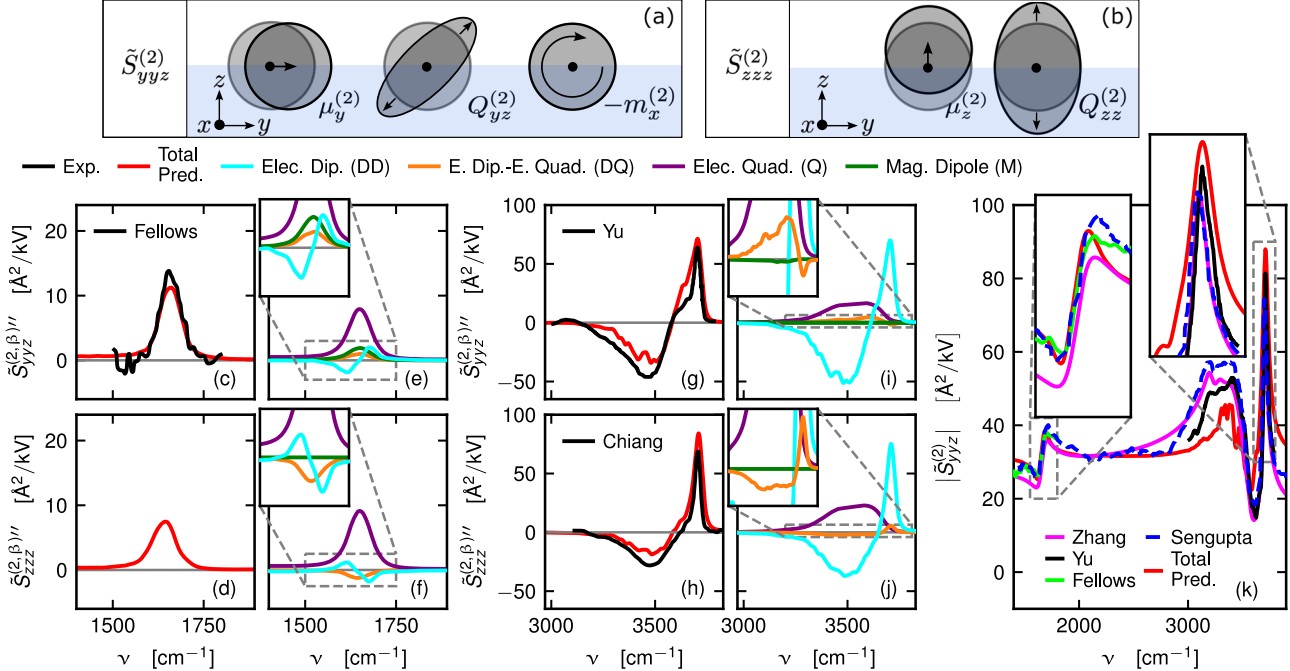

**Fig. 1 | SFG spectra and decomposition into multipole components. a, b** Sketch of the different second-order molecular multipole contributions to the total SFG spectra $\widetilde{S}^{(2)\prime\prime}_{yyz}$ and $\widetilde{S}^{(2)\prime\prime}_{zzz}$, respectively. Here, $\mu^{(2)}_i$, $Q^{(2)}_{ij}$, and $m^{(2)}_i$ denote the induced molecular second-order electric dipole, electric quadrupole, and magnetic dipole moments, respectively. The blue region represents water, and the white region represents air. **c** Comparison of predicted and experimental imaginary SFG spectra $\widetilde{S}^{(2)\prime\prime}_{yyz}$ in the bending-frequency region. The SFG spectrum is decomposed into the pure electric dipole (DD), the electric dipole - electric quadrupole cross (DQ), the electric quadrupole (Q), and the magnetic dipole contribution (M) in (**e**). **d, f** The same analysis for $\widetilde{S}^{(2)\prime\prime}_{zzz}$. Results for the OH-stretch frequency region are shown in (**g–j**). Experimental data are taken from Fellows et al.[54], Yu et al.[64], and Chiang et al.[34]. The predicted absolute spectrum $\left|\widetilde{S}^{(2)}_{yyz}\right|$ is compared with various published experimental data[54,64–66] in (**k**). We red-shift our predictions for the bending and stretch bands by 28 and 166 cm$^{-1}$, respectively, to match the experiments. The boundary between the two shifted regions is set at 2500 cm$^{-1}$. Gray dashed boxes indicate the regions shown in the insets. Source data are provided as a Source data file.

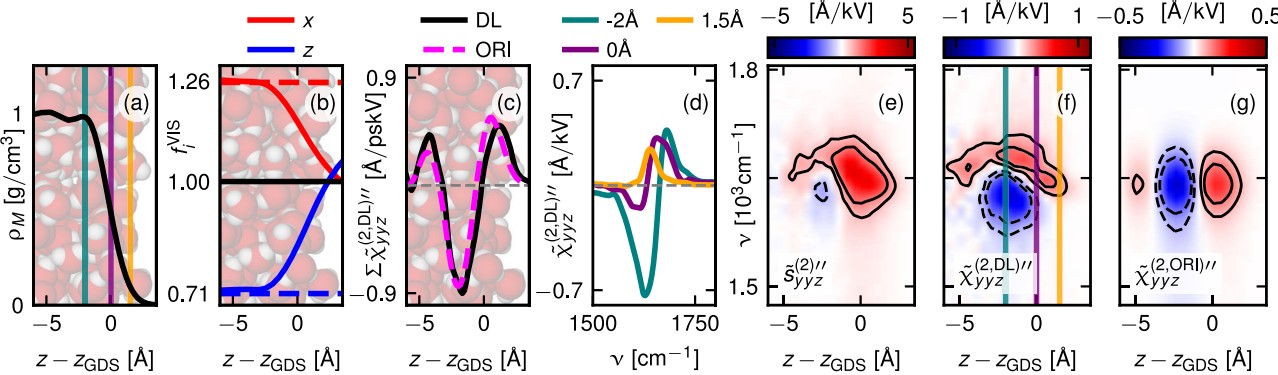

**Fig. 2 | Depth-dependent analysis of the bending band.** The mass density profile is shown in (**a**), the local field factors in the visual frequency range $f_x^{VIS}(z)$ and $f_z^{VIS}(z)$ defined in Eq. (2), in (**b**). Lorentz theory predictions in bulk water are denoted by dashed horizontal lines. The black solid line denotes the vacuum value. For illustrative purposes, a snapshot from the simulation is shown in the background. **c** Comparison of the spatially resolved integrals over the bending band, as defined in Eq. (3), between $\tilde{\chi}_{yyz}^{(2,DL)''}(z)$ and the prediction based solely on molecular orientation, $\tilde{\chi}_{yyz}^{(2,ORI)''}(z)$, defined in Eq. (35). The electric dipole second-order susceptibility $\tilde{\chi}_{yyz}^{(2,DL)''}(z)$, defined in Eq. (30), is shown at selected positions in (**d**). These positions are marked in (**f**), where the corresponding two-dimensional profile is presented. The second-order response profile of the SFG signal $\tilde{s}_{yyz}^{(2)''}(z)$ is presented in (**e**) and $\tilde{\chi}_{yyz}^{(2,ORI)''}(z)$ in (**g**). The spectra are red-shifted by 28 cm$^{-1}$. Source data are provided as a Source data file.

$\tilde{S}_{ijk}^{(2,M)}$ are the electric quadrupole and magnetic dipole contributions, which are in contrast bulk contributions, independent of the interfacial structure. The sum of these contributions $\tilde{S}_{ijk}^{(2,Q)} + \tilde{S}_{ijk}^{(2,M)}$ is called the interfacial quadrupole bulk contribution (IQB) in the literature[45,46].

To give some intuitive understanding of the physical mechanisms producing the different multipolar SFG contributions, we schematically illustrate the second-order molecular multipoles of the valence electron charge cloud in Fig. 1a, b. Figure 1a shows the contributions relevant for a y-polarized and (b) for a z-polarized SFG field. The second-order molecular electric dipole moment $\mu_i^{(2)}$ is characterized by oscillations of the displacement of the center of the electronic charge distribution and is zero in isotropic media. In contrast, the second-order electric quadrupole moment, characterized by oscillations of the charge distribution width, and the molecular magnetic dipole moment, characterized by oscillations in its angular momentum, as illustrated in Fig. 1a, b, are non-zero in bulk. The second-order electric quadrupole and magnetic dipole moments consist of electric current distributions with vanishing mean and contribute to the second-order current density via the gradients of their spatial distributions.

The bending band imaginary SFG spectra in Fig. 1c–f are dominated by the electric quadrupole contribution (purple), which consists of a positive broad band, in agreement with previous normal mode calculations[50]. The pure electric dipole contribution consists of a rather weak negative-positive double peak (Fig. 1e, f). The observed lineshape of $\tilde{S}_{yyz}^{(2,DD)''}$ qualitatively aligns with previous simulation studies[55,56,58], which did not account for higher-order multipole contributions. The magnetic dipole contribution appears only in the *yyz* spectrum and exhibits an intensity similar to the pure electric dipole contribution (Fig. 1e).

The OH-stretch band imaginary SFG spectra are presented in Fig. 1g–j. The pure electric dipole contribution is characterized by a strong negative-positive double peak, in shape similar to the bending band, as can be seen by comparing the cyan lines in Fig. 1e, i. The difference is that the frequency splitting between the positive and negative components is much larger compared to the spectral linewidth in the OH-stretch region, thus less cancellation takes place, and the resulting electric dipole contribution is significantly stronger. The positive signal arising from the free OH stretch vibrations at 3700 cm$^{-1}$

is largely determined by the pure electric dipole contribution (Fig. 1i, j), while significant electric quadrupole contributions to the SFG spectra arise from the more slowly oscillating OH bonds between 3100 and 3650 cm$^{-1}$. In $\tilde{S}_{yyz}^{(2)''}$, the electric quadrupole produces most of the positive shoulder at 3600 cm$^{-1}$ (Fig. 1i). Comparing $\tilde{S}_{yyz}^{(2,DD)''}$ (Fig. 1i) and $\tilde{S}_{zzz}^{(2,DD)''}$ (Fig. 1j), one sees that $\tilde{S}_{yyz}^{(2,DD)''}$ exhibits a small positive shoulder around 3600 cm$^{-1}$, whereas $\tilde{S}_{zzz}^{(2,DD)''}$ does not. This difference is qualitatively consistent with previous predictions, which were based on electric-dipole approximations[34,63,64]. In all spectra presented, $\tilde{S}_{ijk}^{(2,DQ)''}$ is small, and $\tilde{S}_{yyz}^{(2,DQ)''}$ is mainly positive, while $\tilde{S}_{zzz}^{(2,DQ)''}$ is mainly negative.

In Fig. 1k, we compare several experimental measurements of the absolute SFG spectrum $\left|\tilde{S}_{yyz}^{(2)}\right| = \sqrt{\left(\tilde{S}_{yyz}^{(2)'}\right)^2 + \left(\tilde{S}_{yyz}^{(2)''}\right)^2}$[54,64–66] with our prediction (red line), showing close agreement within the spread of experimental data across the full frequency range. This comparison also provides an estimate of the experimental uncertainty.

Our predicted spectra have been red-shifted to ensure alignment between theory and experiment. This is reasonable since it has been demonstrated previously that the MB-Pol water model used by us reproduces the experimental vibrational frequencies when nuclear quantum effects are considered[67–69], while for the linear absorption spectra, it has been demonstrated that nuclear quantum effects do not alter the line shape and absolute intensities much[68,70]. Quantum correction factors do, in fact, not appear in spectra to leading order as discussed in the Methods.

Our results reveal significant higher-order multipole contributions. Next, we show how the interfacial electric-dipole contribution in the bending band reflects molecular orientation at the air–water interface.

## Spatially resolved second-order response profile of the bending band

Figure 2 presents the depth-dependent second-order response profile of the bending band, relative to the Gibbs dividing surface position $z_{GDS}$[71]. Figure 2a presents the mass density profile with a snapshot of the simulation box in the background. The density transitions from 1 g/cm$^3$ in bulk to essentially zero over a range of about 5 Å, centered around $z_{GDS}$.

We present the second-order electric dipole susceptibility $\widetilde{\chi}_{ijk}^{(2,\mathrm{DL})}(z)$, which connects the average local electric E-field at the molecular centers to the resulting second-order electric dipole current density in Fig. 2d, f. The superscript DL stands for dipole and local to distinguish it from regular susceptibilities, which are defined as the response of the macroscopic polarization density to macroscopic electric E-fields[72]. Though not directly experimentally measurable, $\widetilde{\chi}_{ijk}^{(2,\mathrm{DL})}(z)$ is important since it links the macroscopic second-order electric dipole response to molecular orientation[27,28,73]. In defining $\widetilde{\chi}_{ijk}^{(2,\mathrm{DL})}(z)$ via Eq. (30), we account for the strength of the local E-field by dividing the second-order response of the electric dipole density $\widetilde{s}_{ijk}^{(2,\mathrm{DD})}(z)$, by the local field factors, i.e., the ratios between the amplitudes of the local E-fields $\mathcal{E}_i^{\mathrm{L},\alpha}(z)$ acting on the molecular centers and the amplitudes of the external fields $\mathcal{F}_i^\alpha$, which correspond to D-fields for $i = z$ and to E-fields for $i = x$ or $y$. Here, $\alpha$ specifies the frequency of the three fields involved, i.e., $\alpha \in \{\mathrm{SFG, VIS, IR}\}$. The spatial integral over the second-order response profile $\widetilde{s}_{ijk}^{(2,\mathrm{DD})}(z)$ in Eq. (21) determines the pure electric dipole contribution to the experimentally measurable SFG spectrum $\widetilde{S}_{ijk}^{(2,\mathrm{DD})}$; it is related to $\widetilde{\chi}_{ijk}^{(2,\mathrm{DL})}(z)$ by

$$\widetilde{s}_{ijk}^{(2,\mathrm{DD})}(z) = f_i^{\mathrm{SFG}}(z)\, f_j^{\mathrm{VIS}}(z)\, f_k^{\mathrm{IR}}(z)\, \widetilde{\chi}_{ijk}^{(2,\mathrm{DL})}(z)\,, \tag{1}$$

where $f_i^\alpha(z)$ are the laterally averaged local field factors, defined by

$$f_i^\alpha(z) = \frac{\mathcal{E}_i^{\mathrm{L},\alpha}(z)}{\mathcal{F}_i^\alpha(z)}\,. \tag{2}$$

The local field factors in the optical frequency range $f_i^{\mathrm{VIS}}(z)$ are presented in Fig. 2b. Interestingly, $f_x^{\mathrm{VIS}}(z)$ exhibits a line shape similar to the mass density profile, while $f_z^{\mathrm{VIS}}(z)$ is markedly different. In bulk, we recover values dictated by the Lorentz field approximation[28,73] $f_x^{\mathrm{VIS}}(-\infty) \approx (2+\widetilde{\varepsilon}^{\mathrm{VIS}})/3$ and $f_z^{\mathrm{VIS}}(-\infty) \approx (2+\widetilde{\varepsilon}^{\mathrm{VIS}})/(3\widetilde{\varepsilon}^{\mathrm{VIS}})$, as indicated by the horizontal dashed lines. Here, $\widetilde{\varepsilon}^{\mathrm{VIS}} = 1.77$ represents the optical dielectric constant of bulk water, extracted from the plateau value of the dielectric profile presented in Fig. 5b. These findings are qualitatively consistent with the prediction of $f_i^{\mathrm{VIS}}(z)$ by Shiratori and Morita[74].

We compare the imaginary second-order response profile $\widetilde{s}_{yyz}^{(2)\prime\prime}(z)$ defined in Eq. (8) with the second-order electric dipole susceptibility $\widetilde{\chi}_{yyz}^{(2,\mathrm{DL})\prime\prime}(z)$ defined in Eq. (30), as well as its prediction based solely on molecular orientation, $\widetilde{\chi}_{yyz}^{(2,\mathrm{ORI})\prime\prime}(z)$, in Fig. 2e–g. It is evident that all profiles exhibit a positive contribution for values of $z - z_{\mathrm{GDS}} \geq 0$ (in red) and a negative contribution (in blue) centered around $z - z_{\mathrm{GDS}} = -2\,\text{Å}$. We calculate $\widetilde{\chi}_{ijk}^{(2,\mathrm{ORI})}(z)$ by summing the molecular hyperpolarizability tensor contributions assigned to each water molecule in its molecular frame, as defined in Eq. (35), where the molecular hyperpolarizability tensor is extracted from a simulation of bulk water as described in SI Section VII. By this, the only interface-specific input to $\widetilde{\chi}_{ijk}^{(2,\mathrm{ORI})}(z)$ is the molecular orientation distribution, allowing us to test whether $\widetilde{\chi}_{ijk}^{(2,\mathrm{DL})}(z)$ is a marker of the interfacial orientation anisotropy. To compare the amplitudes of $\widetilde{\chi}_{yyz}^{(2,\mathrm{DL})}(z)$ and $\widetilde{\chi}_{yyz}^{(2,\mathrm{ORI})}(z)$ we integrate over the bending band according to

$$\Sigma\widetilde{\chi}_{ijk}^{(2,\mathrm{DL})\prime\prime}(z) = c_0 \int_{\nu_1}^{\nu_2} \mathrm{d}\nu\, \widetilde{\chi}_{ijk}^{(2,\mathrm{DL})\prime\prime}(z,\nu) \tag{3}$$

from $\nu_1 = 1507$ to $\nu_2 = 1772\,\mathrm{cm}^{-1}$. Here, $c_0$ is the speed of light in vacuum. We find quantitative agreement between $\Sigma\widetilde{\chi}_{yyz}^{(2,\mathrm{DL})\prime\prime}(z)$ and $\Sigma\widetilde{\chi}_{yyz}^{(2,\mathrm{ORI})\prime\prime}(z)$ in Fig. 2c, demonstrating that $\Sigma\widetilde{\chi}_{yyz}^{(2,\mathrm{DL})\prime\prime}(z)$ indeed primarily

reflects interfacial orientation within the bending frequency range. The characteristic triple-layer structure of $\Sigma\widetilde{\chi}_{yyz}^{(2,\mathrm{DL})\prime\prime}(z)$ will be discussed further below. By comparing $\widetilde{\chi}_{yyz}^{(2,\mathrm{DL})\prime\prime}(z)$ (Fig. 2d, f) with $\widetilde{\chi}_{yyz}^{(2,\mathrm{ORI})\prime\prime}(z)$ (Fig. 2g), it becomes clear that $\widetilde{\chi}_{yyz}^{(2,\mathrm{ORI})\prime\prime}(z)$ does not reproduce the $z$-dependent frequency shift, present in $\widetilde{\chi}_{yyz}^{(2,\mathrm{DL})\prime\prime}(z)$, which is thus due to the $z$-dependent anisotropic coordination of interfacial water.

## Spectral signature of biaxial interfacial water ordering

After demonstrating that the second-order electric dipole susceptibility $\widetilde{\chi}_{yyz}^{(2,\mathrm{DL})}$ serves as a quantitative probe of the interfacial orientation of molecules, we now analyze the interfacial orientation distribution. The orientation distribution function $\rho_{\mathrm{ORI}}(\theta, \psi)$, defined in Eq. (36), is presented at three depth $z - z_{\mathrm{GDS}}$ in Fig. 3a–c. This function depends on the two Euler angles, $\theta$ and $\psi$. The angle $\theta$ describes the tilt of the molecular dipole axis relative to the surface normal: for $\theta = 0$, the oxygen atom is oriented towards bulk water, and for $\theta = 180°$ towards air. The angle $\psi$ describes the molecular orientation around the dipole axis. The interfacial water structure is significantly biaxial, as revealed by the pronounced $\psi$-dependence of $\rho_{\mathrm{ORI}}(\theta, \psi)$.

We distinguish three orientational species: the planar-oriented species (orange), where both OH-bond vectors lie in the interfacial plane, the inward-oriented species (magenta), where one OH-bond is orthogonal to the interfacial plane and pointing inward, and the outward-oriented species (cyan), where one OH-bond is orthogonal to the interfacial plane and pointing outward. These orientations are sketched on top of Fig. 3d, and the corresponding Euler angles are marked with color-coded dots and contours (marking water molecules with an angular deviation of 36° from the idealized orientations) in the plots of $\rho_{\mathrm{ORI}}(\theta, \psi)$ in Fig. 3a–c. A snapshot of the air–water interface viewed from air is presented in Fig. 3d; here, the orientational species are color-coded.

As observed in Fig. 3a, at $z - z_{\mathrm{GDS}} = -2\,\text{Å}$, the inward-oriented species dominates; at $z = z_{\mathrm{GDS}}$, the planar species prevails (Fig. 3b), and finally, close to the vapor region at $z - z_{\mathrm{GDS}} = 1.5\,\text{Å}$, the outward-oriented species becomes dominant (Fig. 3c). We quantify this observation via the molecular number fraction profiles, which we present in Fig. 3e. Here, we see that water is isotropic $5\,\text{Å}$ below $z_{\mathrm{GDS}}$, as the fraction of all orientational species is approximately 20%. The inward-oriented fraction peaks at $z - z_{\mathrm{GDS}} = -2\,\text{Å}$, at the location of the negative peak of $\widetilde{\chi}_{ijk}^{(2,\mathrm{DL})}(z)$ in Fig. 2f. The planar-oriented and outward-oriented species peak at $z = z_{\mathrm{GDS}}$ and near the vapor region at $z - z_{\mathrm{GDS}} = 2.5\,\text{Å}$, respectively.

The uniaxial approximation, which corresponds to averaging the orientation distribution function over the angle $\psi$, neglects the biaxial water ordering. To reveal the spectral effects of water biaxial ordering, we define the orientation and position dependent molecular hyperpolarizability $\widetilde{\beta}_{ijk}(z, \theta, \psi)$ by

$$\varepsilon_0^{-1}\mu_i^{(2)}(t, z, \theta, \psi) = e^{-i\omega^{\mathrm{SFG}}t}\,\widetilde{\beta}_{ijk}(z, \theta, \psi)\,\mathcal{E}_j^{\mathrm{L,VIS}}(z)\,\mathcal{E}_k^{\mathrm{L,IR}}(z) + c.c.\,, \tag{4}$$

where $\mu_i^{(2)}(t)$ is the second-order molecular electric dipole moment, $\mathcal{E}_j^{\mathrm{L,VIS}}(z)$ and $\mathcal{E}_k^{\mathrm{L,IR}}(z)$ are the amplitudes of the local E-fields acting on the molecular center at the position $z$, and $\theta$ and $\psi$ are the Euler angles, specifying the molecular orientation and the Einstein summation convention is used. Note that $\mu_i^{(2)}(t)$ is a source dipole that induces an additional linear response, as explained in SI Section IV B. We extract $\widetilde{\beta}_{yyz}(\theta, \psi)$, from simulation trajectories of bulk water, as described in SI Section VII. In Fig. 3f, we present the frequency integral of the imaginary part $\widetilde{\beta}_{yyz}^{\prime\prime}(\theta, \psi)$ over the same integration boundaries as in Eq. (3) to quantify the orientation-dependent hyperpolarizability in the bending band. We decompose $\widetilde{\beta}_{yyz}(\theta, \psi)$ into uniaxial and biaxial

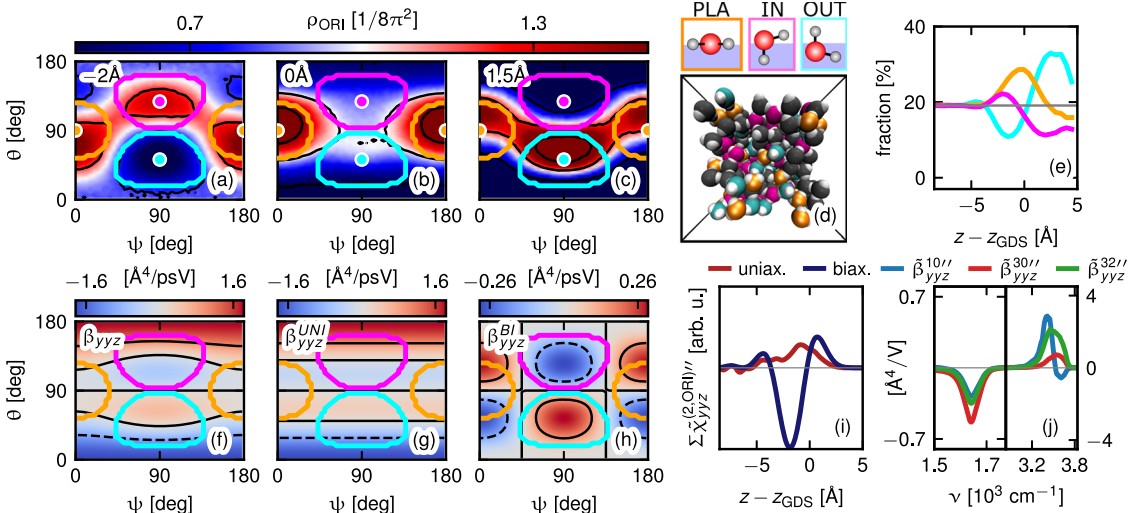

**Fig. 3 | Orientational analysis of the bending band.** The orientation distribution function $\rho_{ORI}(\theta, \psi)$ is shown for the same positions relative to $z_{GDS}$ as in Fig. 2 in (a–c). Three orientational species are distinguished: planar, pointing in, and pointing out, as illustrated at the top of (d). These orientations are marked with dots and contours in (a–c). Additionally, a snapshot of the simulation box viewed from the air is provided in (d), where molecules are color-coded according to their orientational species. **e** The fraction of each orientational species as a function of $z$. The molecular hyperpolarizability tensor component $\tilde{\beta}_{yyz}(\theta, \psi)$, defined in Eq. (4) and obtained from bulk simulation, is shown as a function of the molecular orientation in (f). The color indicates the intensity at bending-band frequencies, obtained by integrating the imaginary part $\tilde{\beta}''_{yyz}(\theta, \psi)$ over frequency using the same integration boundaries as in Eq. (3). This intensity is decomposed into its uniaxial and biaxial components, defined in Eqs. (6) and (7) and shown in (g) and (h), respectively. The position-resolved profile of the bending band is decomposed into uniaxial and biaxial contributions in (i). The frequency-dependent coefficients $\tilde{\beta}^{lm''}_{yyz}$ of $\tilde{\chi}^{(2, ORI)''}_{yyz}$ defined in Eqs. (5)–(7) are presented in (j). Source data are provided as a Source data file.

components according to

$$\tilde{\beta}_{ijk}(\theta, \psi) = \tilde{\beta}^{UNI}_{ijk}(\theta) + \tilde{\beta}^{BI}_{ijk}(\theta, \psi) \tag{5}$$

$$\tilde{\beta}^{UNI}_{ijk}(\theta) = q_{10}(\theta)\tilde{\beta}^{10}_{ijk} + q_{30}(\theta)\tilde{\beta}^{30}_{ijk} \tag{6}$$

$$\tilde{\beta}^{BI}_{ijk}(\theta, \psi) = q_{32}(\theta, \psi)\tilde{\beta}^{32}_{ijk} \tag{7}$$

shown in Fig. 3g, h, respectively. Here, $q_{10}(\theta) = \cos\theta$, $q_{30}(\theta) = \frac{1}{2}[5\cos^3\theta - 3\cos\theta]$ are the first and third Legendre polynomials specifying the dipole distribution, and $q_{32}(\theta, \psi) = \cos\theta \sin\theta^2 (\cos^2\psi - \sin^2\psi)$ accounts for the contribution due to the molecular biaxiality. Equations (5)–(7) express the rotation of the molecular hyperpolarizability into the laboratory frame in an exact way. The frequency-dependent imaginary parts of the coefficients $\tilde{\beta}^{10}_{yyz}$, $\tilde{\beta}^{30}_{yyz}$, and $\tilde{\beta}^{32}_{yyz}$ are presented in Fig. 3j, where we observe that the biaxial contribution $\tilde{\beta}^{32}_{yyz}$ is significant.

We observe that the negative area (blue) at $\theta > 90°$ in $\tilde{\beta}_{yyz}(\theta, \psi)$ in Fig. 3f aligns with the inward pointing orientation dominant at $-2$ Å in Fig. 3a and the positive area (red) at $\theta < 90°$ in Fig. 3f aligns with the outward pointing orientation dominant at 1.5 Å in Fig. 3c. As seen in Fig. 3g, the uniaxial contribution changes sign within the domain of each orientational species, suggesting that uniaxial water ordering by itself does not lead to a SFG signal in the bending region, as will be rigorously demonstrated below. In contrast, the biaxial component precisely projects out the difference between outward- and inward-oriented molecules (Fig. 3h). We note that planar-oriented molecules are not detected by SFG spectroscopy (neither uniaxial nor biaxial), because their mirror plane lies parallel to the interface, making them inversion symmetric. We dissect the integral over the bending band $\Sigma\tilde{\chi}^{(2, ORI)''}_{yyz}(z)$, already presented in Fig. 2c, into biaxial and uniaxial contributions in Fig. 3i, demonstrating that $\Sigma\tilde{\chi}^{(2, ORI)''}_{yyz}(z)$ is dominated

by the biaxiality of the orientation distribution. Thus, our analysis of the simulation trajectories shows that interfacial water is significantly biaxial and that the sign of the bending-mode dipole response is an unambiguous indicator of the relative populations of biaxially ordered inward- and outward-oriented water molecules illustrated in Fig. 3d.

## Spatially resolved second-order response profile of the stretch band

We present the imaginary part of the second-order response profile $\tilde{s}^{(2)''}_{ijk}(z)$ in the OH-stretch frequency region, defined in Eq. (8), together with a decomposition into the electric dipole and electric quadrupole contributions according to Eq. (20) in Fig. 4. We map the waters center of mass positions in the calculation of $\tilde{s}^{(2)}_{ijk}(z)$ relative to the non-planar Willard-Chandler surface $z_{WCS}$[75] in Fig. 4e–h and relative to the planar Gibbs dividing surface $z_{GDS}$ in Fig. 4c, d, i–n. We show a snapshot of the simulation box in Fig. 4a. In Fig. 4b, we present the mass-density profiles relative to $z_{GDS}$ and $z_{WCS}$. In the latter presentation, the density profile strongly oscillates, while in the laboratory frame, these oscillations are washed out due to the intrinsic roughness of the air–water interface[76].

Slices of $\tilde{s}^{(2)''}_{ijk}(z)$ at selected positions $z - z_{GDS}$ and $z - z_{WCS}$ (denoted by the colored lines in (b)) are presented in Fig. 4c–f. Close to the vapor at $z - z_{GDS/WCS} = 0.5$ Å (orange lines), almost only the free OH stretch peak is visible at 3700 cm⁻¹. At $z - z_{GDS/WCS} = -0.5$ Å (purple lines), $\tilde{s}^{(2)''}_{ijk}(z)$ resembles the integrated spectra presented in Fig. 1g, h. Finally, the free OH stretch contributions vanish near the bulk at $z - z_{GDS/WCS} = -1.5$ Å (turquoise lines), and we only see the shoulder at 3600 cm⁻¹ and the negative component at 3500 cm⁻¹. As shown in Fig. 4h, $\tilde{s}^{(2)''}_{zzz}(z)$ relative to $z_{WCS}$ almost perfectly follows the oscillations of the mass density profile in Fig. 4b.

The imaginary part of the second-order response profile in the laboratory frame $\tilde{s}^{(2)''}_{ijk}(z)$ is presented in Fig. 4c, d, i, j. Here, both the free OH stretch band at 3700 cm⁻¹ and the negative contribution at 3500 cm⁻¹ are located at approximately $z \approx z_{GDS}$. However, the shoulder is created below $z_{GDS}$, and the free-OH contributions reach slightly more into the vapor region. The profiles of the second-order response

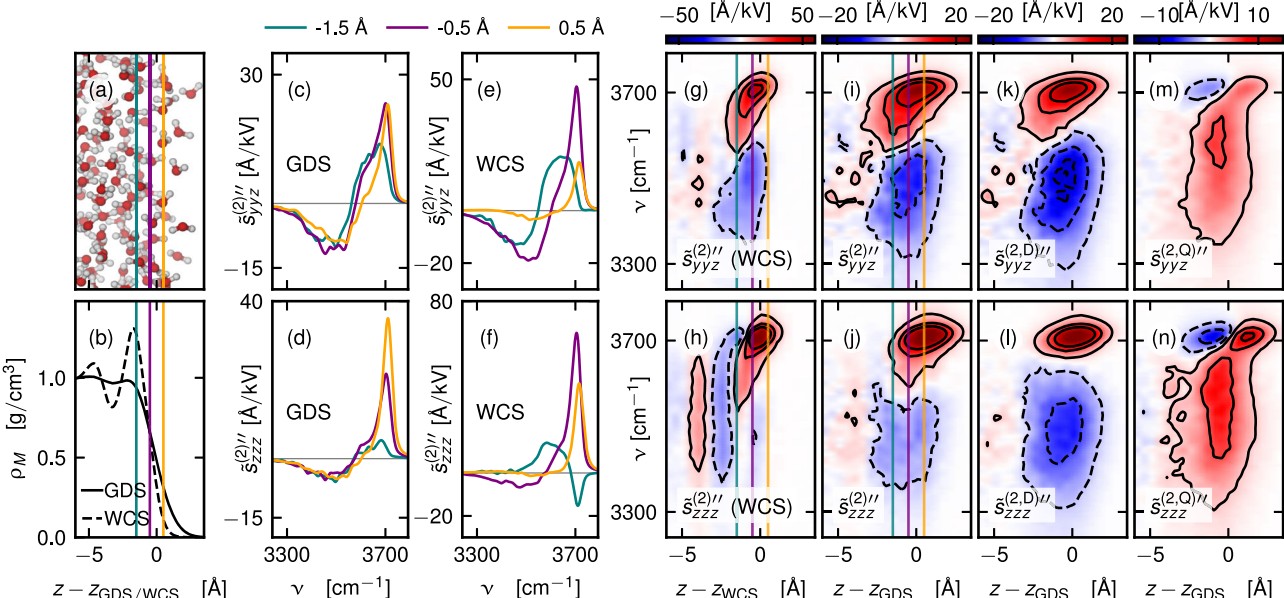

**Fig. 4 | Depth-dependent analysis of the stretching band.** Second-order response profile $\tilde{s}_{ijk}^{(2)\prime\prime}(z)$ of the OH-stretch band as defined in Eq. (8) and the decomposition into its multipole components $\tilde{s}_{ijk}^{(2,\beta)\prime\prime}(z)$ as defined in Eq. (20). A snapshot of the simulation box is shown in (**a**). The mass-density profiles with respect to $z_{GDS}$ and $z_{WCS}$ are presented in (**b**). Slices of $\tilde{s}_{ijk}^{(2)\prime\prime}(z)$ are shown for selected positions in (**c**, **d**) relative to $z_{GDS}$ and in (**e**, **f**) relative to $z_{WCS}$, colored vertical lines in (**a**, **b**, **g**–**j**) indicate the positions. The profiles $\tilde{s}_{ijk}^{(2)\prime\prime}(z)$ are shown relative to $z_{WCS}$ in (**g**, **h**) and relative to $z_{GDS}$ in (**i**, **j**). The profile relative to $z_{GDS}$ is dissected into its molecular multipole components $\tilde{s}_{ijk}^{(2,D)\prime\prime}(z)$ and $\tilde{s}_{ijk}^{(2,Q)\prime\prime}(z)$ in (**k**), (**l**), (**m**), and (**n**), respectively. All spectra are red-shifted by 166 cm$^{-1}$. Source data are provided as a Source data file.

of the electric dipole contribution $\tilde{s}_{ijk}^{(2,D)\prime\prime}(z)$ are presented in Fig. 4k, l. These qualitatively agree with previously published depth-resolved SFG spectra[24,63,77–81].

The second-order electric dipole contributions to the SFG signal in the stretching band (Fig. 4k, l) and to the susceptibility in the bending band (Fig. 2f) undergo significant frequency shifts as a function of position $z$ in the interfacial layer. As shown elsewhere[82], when going from vapor to liquid, frequency shifts arise from competing effects of non-Markovian friction (which causes blue-shifting) and potential broadening (which causes red-shifting). The former dominates in the bending band, as seen in Fig. 2f, while the latter dominates in the stretch band, as seen in Fig. 4k, l.

The electric quadrupole profile $\tilde{s}_{ijk}^{(2,Q)\prime\prime}(z)$ is presented in Fig. 4m, n, and is dominated by a broad positive peak around $z \approx z_{GDS}$. Additionally, we observe a negative contribution closer to the bulk region and a positive contribution closer to the vapor region at the frequency of the free OH vibrations. It is important to note that the electric quadrupole profile $\tilde{s}_{ijk}^{(2,Q)\prime\prime}(z)$ strongly depends on position. Consequently, the electric quadrupole contribution to the SFG spectra, $\tilde{S}_{ijk}^{(2,Q)}$ as defined in Eq. (21), could be used to report on interfacial water structure for a non-vanishing wave vector mismatch $\Delta k_z$, which is an interesting venue for future experimental investigation.

While the bending band probes mostly the biaxial interfacial water ordering, as shown in the section "Results" "Spectral signature of biaxial interfacial water ordering," the stretch band is instead more sensitive to the local hydrogen-bonding environment, which varies rather abruptly across the interface[34,83]. Therefore, the second-order electric-dipole profile of the stretch band is confined to a narrow region around the Gibbs dividing surface, whereas the bending profile extends slightly deeper into the bulk, as seen by comparing Fig. 4k with Fig. 2f.

## Spatially resolved linear dielectric and absorption profile

As mentioned before, the $z$-independent external field amplitude $\mathcal{F}_z^\alpha$ corresponds to the D-field, while $\mathcal{F}_{x/y}^\alpha$ corresponds to the E-field. The

relationship between D- and E-fields is determined by the dielectric profile[84–90]. The real parts of the dielectric profiles $\tilde{\varepsilon}_{xx}^{VIS\prime}(z)$ (red) and $\tilde{\varepsilon}_{zz}^{VIS\prime}(z)$ (blue) at optical frequencies are presented in Fig. 5b and have an almost identical shape to the mass density profile (black). Thus, in contrast to the static or THz case[85,91], the anisotropy of the tensorial interfacial response for optical frequencies is rather small, as demonstrated in the inset. The bulk plateau value of 1.77 agrees well with the experimental value of 1.78 at room temperature under atmospheric pressure[92].

We present the imaginary part of the position and frequency-dependent dielectric profiles, $\tilde{\varepsilon}_{xx}^{IR\prime\prime}(z)$ and $\tilde{\varepsilon}_{zz}^{IR\prime\prime}(z)$ extracted from molecular dynamics simulations using Eqs. (27) and (28) in Fig. 5c, d. We observe that the dielectric profile $\tilde{\varepsilon}_{ij}^{IR\prime\prime}(z)$ varies drastically in the interface region and is significantly anisotropic. We see that the parallel component of the dielectric profile $\tilde{\varepsilon}_{xx}^{IR\prime\prime}(z)$ (Fig. 5c, e, h) is characterized by an almost frequency-independent decrease of intensity across the interface, without significant changes in the line shape.

The perpendicular profile $\tilde{\varepsilon}_{zz}^{IR\prime\prime}(z)$ (Fig. 5d, f, i) is more complex. The OH-stretch band, which is in bulk centered around 3500 cm$^{-1}$, exhibits a blue shift when going from bulk to vapor until only the free OH-stretch contribution at 3700 cm$^{-1}$ survives. The libration band[93,94], centered around 500 cm$^{-1}$, is more pronounced in $\tilde{\varepsilon}_{zz}^{IR\prime\prime}(z)$ than in $\tilde{\varepsilon}_{xx}^{IR\prime\prime}(z)$, as can be seen in Fig. 5e, f. This can be rationalized by the higher concentration of planar-oriented molecules near the interface (Fig. 3e), which are more susceptible to $z$-polarized fields.

We observe that isotropically averaged interfacial and bulk spectra are indistinguishable at $z - z_{GDS} = -5.5$ Å in Fig. 5g, j. Hence, the second-order response in Figs. 2 and 4, as well as the linear response in Fig. 5, and the orientational anisotropy in Fig. 3, can be considered bulk-like at this depth. The comparison with experiments[95] (black) in Fig. 5g, j shows that the differences between simulation and experiments are rather small around the bending peak and in the OH-stretch region at high frequencies ($>3500$ cm$^{-1}$) where the SFG stretch signal is most intense, whereas the experimental IR intensity is significantly underestimated at lower frequencies.

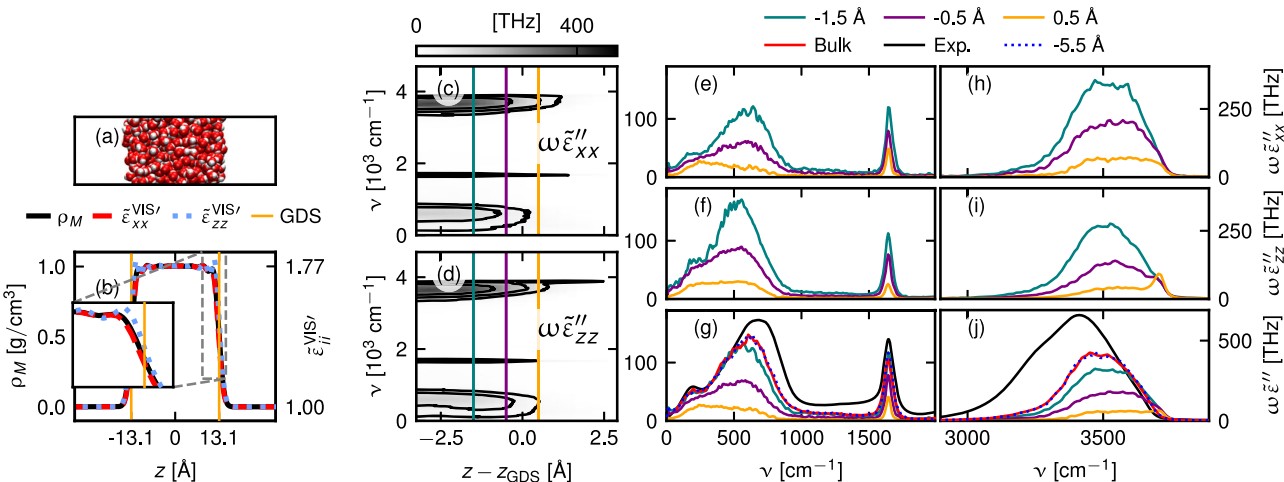

**Fig. 5 | Linear optical dielectric and IR absorption profiles at the air–water interface. a** Snapshot of the simulation. **b** The water mass density profile is compared to the real parts of the tensorial optical dielectric profiles. **c, d** The linear absorption profiles $\omega \widetilde{\varepsilon}''_{xx}(z)$ and $\omega \widetilde{\varepsilon}''_{zz}(z)$ in the IR and low THz frequency range are shown as a 2D plot. **e–j** The imaginary part of the linear absorption profile is shown at the same positions relative to the Gibbs dividing surface as in Fig. 4. The $xx$-component defined in Eq. (28), the $zz$-component defined in Eq. (27), and the isotropic average defined in Eq. (29) are presented in (**e**), (**f**), and (**g**), respectively. Additionally, the predicted linear absorption spectrum of bulk water and the experimental spectrum[95] are shown in (**g**). **h–j** Results in the stretch region. The bending band in (**e–g**) is redshifted by 28 cm$^{-1}$ and the stretching band in (**h–j**) is redshifted by 166 cm$^{-1}$. Source data are provided as a Source data file.

## Discussion

Accounting for higher-order multipole contributions is essential for interpreting SFG spectra and leads to quantitative agreement between simulations and experimental data. In the bending band, the electric dipole contribution is almost negligible due to the small frequency shift between mutually canceling positive and negative SFG signals from water at different depths, and has an amplitude comparable to the usually neglected magnetic dipole contribution. As a result, the total SFG signal is dominated by the electric quadrupole contribution. The shoulder appearing at 3600 cm$^{-1}$ in the stretch band has been attributed to orientational anisotropy[96,97], or to a combination mode involving intermolecular coupling[98], as summarized in the review by Tang et al.[99]. Our results demonstrate that the shoulder primarily originates from electric quadrupole contributions, which do not reflect interfacial structure but rather depend on bulk properties for vanishing wave vector mismatch $\Delta k_z = 0$. Only after subtracting higher-order multipoles can the anisotropic orientation of interfacial molecules be inferred from the SFG signal, by which pronounced biaxial water ordering at the air–water interface is revealed. This biaxial structure consists of essentially three layers of orientational species, namely an inwards-oriented, a planar-oriented, and an outwards-oriented species. This arrangement is surprisingly similar to structural motifs observed at the air-ice interface[100].

In conclusion, since SFG spectra are significantly influenced by non-interface-specific multipole effects, it is crucial to account for and subtract these contributions in order to be able to interpret experimental spectra in terms of interfacial orientations. This can be achieved by subtracting a reference spectrum with the same bulk medium[54], or by using theoretical predictions, as those provided in this work.

## Methods
### Technical details

All simulations are done in the NVT ensemble at room temperature $T = 298$ K with the MB-Pol force field[101–103]. For starting configurations, 94 equally spaced initial configurations are extracted from a 18.8 ns long simulation using the SPC/E force field[104] and the GROMACS molecular dynamics software[105] (Version 2019.3). The SPC/E simulations are done with a time step of 2 fs and a velocity rescaling thermostat[106], with a relaxation time of 1 ps. Each of the initial configurations is equilibrated with the MB-Pol force field for 20 ps, and afterward, production runs with an average runtime of 0.86 ns are executed using the LAMMPS software[107] (Version 23 Jun 2022, Update 1). Again, a velocity-rescaling thermostat is employed. However, for MB-Pol simulations, the relaxation time is set to 5 ps, and the time step is 0.2 fs. The box size is 2 nm in the $x$ and $y$ dimensions and 6 nm in the $z$ dimension. This size is chosen to ensure that the box contains more than twice as much air as water, since the water slab spans approximately 2.6 nm. The box is filled with 352 water molecules, and the two interfaces lie in the $xy$ plane.

The total dipole moment of the simulation box is computed with the modified TTMF-4 model, which describes the electrostatics of the MB-Pol potential[70,101]. The same model is used to predict the linear absorption profile, where we compute polarization density trajectories from monopole and dipole density trajectories. Molecular polarizabilities are computed with single-molecule quantum chemistry calculations using the software Gaussian 16[108]. The theoretical level is CCSD(T)/aug-cc-pVTZ and B3LYP/aug-cc-pVTZ in the parameterization of $\alpha_{ij}^{n,\mathrm{DD}}(t)$, and $\alpha_{ijk}^{n,\mathrm{QD}}(t)$, appearing in Eqs. (22) and (23), respectively. Electrostatic interactions between the induced multipoles are computed as described in SI Sections V B and VIII, involving the Ewald summation algorithm included in OpenMM[109].

The simulation of bulk water is done in a cubic box with a length of 2.07 nm, consisting of 297 molecules, which corresponds to the density in the liquid phase of the slab system. The initial configuration for the MB-Pol simulation used to extract the bulk dielectric constant and the molecular hyperpolarizability was taken from a trajectory generated with the SPC/E model after a simulation time of 20 ns. The magnetic dipole contribution is extracted from a different trajectory, because it requires a smaller write-out frequency. For the calculation of the magnetic dipole contribution, the initial configuration for the MB-Pol simulation was taken from a trajectory generated with the SPC/E model after a simulation time of 40 ns. Both trajectories were produced using the same simulation parameters as described above, but with GROMACS version 2023.3. The first 10 ps of the MB-Pol trajectory is discarded, and a trajectory of length of 400 ps is used to calculate the bulk dielectric constant and the molecular hyperpolarizability, and a trajectory of length of 300 ps is used to calculate the second-order

magnetic dipole susceptibility. Additional MB-Pol simulations are generated to compute the SFG signal from the isotropic interface presented in Supplementary Fig. 9. For these results, we averaged over eight trajectories, each of a length of 300 ps, with initial configurations taken from the same bulk SPC/E trajectory at times 2, 4, 6, 8, 10, 12, 14, 18 ns. The simulation parameters of the bulk simulations are the same as for the interface. We use the recommended settings of the MB-Pol potential from the GitHub repository "MBX - Version 0.7" from the Paesani group[110]. The write-out frequency is 3.2 fs for all spectra except 0.4 fs for the magnetic dipole contribution. In all simulations, periodic boundary conditions are imposed. However, we correct for the external field modification arising from the periodic replicas as described in SI Section IX. The smoothing procedure and the error estimation method of the predicted spectra are explained in SI Section X. Details on the comparison between the experimental and simulated absolute spectra presented in Fig. 1k are given in SI Section XI.

## Prediction of locally resolved SFG spectra

We consider a planar interface between two media. The system is homogeneous in the $xy$ plane but inhomogeneous along the $z$ dimension. The system is illuminated by two monochromatic light sources with frequencies $\omega^{VIS}$ and $\omega^{IR}$ that induce an electric current density $j_i^{(2)}(z, t)$, oscillating with the sum frequency $\omega^{SFG} = \omega^{IR} + \omega^{VIS}$. Here, $\omega^{IR}$ is in the IR range and invokes equifrequent oscillation of the nuclei, while $\omega^{VIS}$ is in the optical frequency range. The resulting second-order electric current density is determined by the second-order response of a time-dependent perturbation expansion[61]

$$\varepsilon_0^{-1} j_i^{(2)}(z, t) = -i\omega^{SFG} e^{-i\omega^{SFG}t} \tilde{s}_{ijk}^{(2)}(z) \mathcal{F}_j^{VIS} \mathcal{F}_k^{IR} + c.c., \quad (8)$$

where $i, j, k \in \{x, y, z\}$ are Cartesian coordinate indices and the Einstein summation convention is used. Here $\tilde{s}_{ijk}^{(2)}(z)$ is the second-order response profile, the tilde denotes a Fourier transformation, i.e., $\tilde{\phi}(\omega) = \mathrm{FT}[\phi(t)](\omega)$, $\tilde{\phi}'$ and $\tilde{\phi}''$ stand for the real and imaginary parts, and $c.c.$ denotes the complex conjugate. The external field amplitude $\mathcal{F}_i^{\alpha}$ corresponds to[84–88]

$$\mathcal{F}_i^{\alpha} = \left(\delta_{ix} + \delta_{iy}\right) \mathcal{E}_i^{\alpha} + \delta_{iz} \varepsilon_0^{-1} \mathcal{D}_i^{\alpha}, \quad (9)$$

where $\mathcal{E}_i^{\alpha}$ and $\mathcal{D}_i^{\alpha}$ are the amplitudes of the E-fields and the D-fields

$$E_i^{\alpha}(z, t) = \mathcal{E}_i^{\alpha}(z) e^{-i\omega^{\alpha}t} + c.c. \quad (10)$$

$$D_i^{\alpha}(z, t) = \mathcal{D}_i^{\alpha}(z) e^{-i\omega^{\alpha}t} + c.c., \quad (11)$$

oscillating with the frequency $\omega^{\alpha}$, $\alpha \in \{SFG, VIS, IR\}$ and $\delta_{ij}$ is the Kronecker delta. Hence, the symbol $\mathcal{F}_i^{\alpha}$ is a placeholder for the spatially constant amplitude of D-fields or E-fields on the relevant length scale, depending on the polarization. The external fields are given by $F_i^{\alpha}(t) = \mathcal{F}_i^{\alpha} e^{-i\omega^{\alpha}t} + c.c.$. By formulating the theory with respect to these fields, we avoid locality approximations. We note that $\tilde{s}_{ijk}^{(2)}(z)$ is distinct from the second-order susceptibility $\tilde{\chi}_{ijk}^{(2)}(z)$, which defines response to electric E-fields.

Several studies have defined $\tilde{\chi}_{ijk}^{(2)}(z)$ using constitutive relations that involve a position-independent effective interfacial dielectric constant $\tilde{\varepsilon}_{eff}^{\alpha}$ [34,64,66]. Within that framework $\tilde{s}_{ijk}^{(2)}(z)$ and $\tilde{\chi}_{ijk}^{(2)}(z)$ are related by

$$\tilde{\chi}_{yyz}^{(2)}(z) = \tilde{\varepsilon}_{eff}^{IR} \tilde{s}_{yyz}^{(2)}(z) \quad (12)$$

$$\tilde{\chi}_{zzz}^{(2)}(z) = \tilde{\varepsilon}_{eff}^{SFG} \tilde{\varepsilon}_{eff}^{VIS} \tilde{\varepsilon}_{eff}^{IR} \tilde{s}_{zzz}^{(2)}(z), \quad (13)$$

with similar relations for other tensor elements. The relation in Eq. (1) used by us is exact on the linear response level and has the advantage that it fully captures position-dependent dielectric effects.

In this work, we only consider the signal arising from the interface on the length scale at which the external fields are spatially constant; therefore, we neglect contributions induced by external field gradients. These are so-called bulk quadrupole contributions[28,45,46], which can be experimentally estimated using combined transmission-reflection SFG[111], combined SFG/DFG techniques[24,54], or configuration analysis and are discussed in SI Section IV C.

It can be shown that $\tilde{S}_{yyz}^{(2)}$ is largely insensitive to bulk quadrupole contributions, provided that the transmitted-beam wavevectors remain approximately parallel[45]. Therefore, $\tilde{S}_{yyz}^{(2)}$ presented in Fig. 1c, g, k can be assumed to be not significantly perturbed by bulk quadrupole contributions. However, this is not the case for $\tilde{S}_{zzz}^{(2)}$ presented in Fig. 1d, h. We compare our theoretical prediction with a previous experimental configuration analysis[112] in SI Section XI and thereby confirm that our comparison is not significantly affected by bulk quadrupole contributions.

The SFG spectrum follows by integration of the second-order response profile over $z$ as

$$\tilde{S}_{ijk}^{(2)} = \int_{-\infty}^{\infty} dz \, e^{-i\Delta k_z z} \tilde{s}_{ijk}^{(2)}(z), \quad (14)$$

where $\Delta k_z$ is the wave vector mismatch defined in SI Section II. We emphasize that Eq. (14) holds whenever the external fields can be assumed to be constant in the region where the dielectric profile is inhomogeneous. This assumption is well satisfied in our case, because the shortest wavelength considered is $\lambda^{SFG} \approx 600$ nm, and the region where the dielectric profile is inhomogeneous is only a few angstroms thick, as demonstrated in this work. Here we utilize the additional approximation $e^{-i\Delta k_z z} \tilde{s}_{ijk}^{(2)}(z) \approx \tilde{s}_{ijk}^{(2)}(z)$. At the air water interface, this is a good approximation, as a typical value for $\Delta k_z$ is 0.02 nm$^{-1}$, whereas the thickness of the air–water interface is about 8 Å[24]. A derivation of Eqs. (8)–(14) is provided in SI Sections I and II.

## Molecular multipole contributions

The second-order electric current density in Eq. (8) can be decomposed into molecular multipole contributions according to

$$j_i^{(2)}(z, t) = j_i^{(2,D)}(z, t) + j_i^{(2,Q)}(z, t) + j_i^{(2,M)}(z, t) + \ldots, \quad (15)$$

where $D$, $Q$, and $M$ stand for electric dipole, electric quadrupole, and magnetic dipole, respectively. The multipole contributions to the second-order current density are defined as $j_i^{(2,D)}(z, t) = \frac{\partial}{\partial t} \rho_i^{(2,D)}(z, t)$, $j_i^{(2,Q)}(z, t) = -\frac{\partial}{\partial t} \frac{\partial}{\partial r_j} \rho_{ij}^{(2,Q)}(z, t)$, and $j_i^{(2,M)}(z, t) = \varepsilon_{ijk} \frac{\partial}{\partial r_j} m_k^{(2)}(z, t)$, where $\rho_i^{(2,D)}(z, t)$ and $\rho_{ij}^{(2,Q)}(z, t)$ are the electric dipole and electric quadrupole densities, $r_i$ is the $i$th Cartesian coordinate of a vector position, $m_i^{(2)}(z, t)$ is the magnetic dipole density and $\varepsilon_{ijk}$ is the Levi-Civita symbol[113,114]. Higher-order molecular multipole contributions in Eq. (15) do not contribute to the experimentally detectable SFG spectrum $\tilde{S}_{ijk}^{(2)}$ in Eq. (14) in the limit $z\Delta k_z \to 0$ and consequently need not be considered[44].

Using a timescale-separation approximation, the second-order electric dipole density can be dissected into the contributions due to source electric dipole $\rho^{DS}(\vec{r}, t)$ and source quadrupole $\rho_{ij}^{QS}(\vec{r}, t)$ densities. Specifically, $\rho_i^{(2,D)}(z, t)$ can be divided into the pure electric dipole contribution $\rho_i^{(2,DD)}(z, t)$ induced by $\rho^{DS}(\vec{r}, t)$, and the electric dipole - electric quadrupole cross contribution $\rho_i^{(2,DQ)}(z, t)$, induced by $\rho_{ij}^{QS}(\vec{r}, t)$. As shown in SI Section V B, these contributions can be defined

by

$$\rho_i^{(2,\mathrm{DD})}(z,t) = \frac{1}{A}\int \mathrm{d}x \int \mathrm{d}y \left[\rho_i^{\mathrm{DS}}(\vec{r},t) + \int \mathrm{d}\vec{r}'\varepsilon_0\tilde{s}_{ij}^{\mathrm{NL}}(\vec{r},\vec{r}',t)F_j^{\mathrm{DS}}(\vec{r}',t)\right]$$

(16)

and

$$\rho_i^{(2,\mathrm{DQ})}(z,t) = \frac{1}{A}\int \mathrm{d}x \int \mathrm{d}y \int \mathrm{d}\vec{r}'\varepsilon_0\tilde{s}_{ij}^{\mathrm{NL}}(\vec{r},\vec{r}',t)F_j^{\mathrm{QS}}(\vec{r}',t),$$

(17)

where $F_i^{\mathrm{DS}}(\vec{r},t)$ and $F_i^{\mathrm{QS}}(\vec{r},t)$ are the electrostatic fields created by the densities $\rho_i^{\mathrm{DS}}(\vec{r},t)$ and $\rho_{ij}^{\mathrm{QS}}(\vec{r},t)$, $A$ is the interfacial area and $\tilde{s}_{ij}^{\mathrm{NL}}(\vec{r},\vec{r}',t)$ is a nonlocal, linear and instantaneous response function. We note that the decomposition $\rho_i^{(2,\mathrm{D})}(z,t)=\rho_i^{(2,\mathrm{DD})}(z,t)+\rho_i^{(2,\mathrm{DQ})}(z,t)$ does not enter the calculation of $\tilde{s}_{ijk}^{(2)}$ and is only used for the interpretation of the SFG spectra. We define the corresponding electric current densities $j_i^{(2,\mathrm{DD})}(z,t)=\frac{\partial}{\partial t}\rho_i^{(2,\mathrm{DD})}(z,t)$ and $j_i^{(2,\mathrm{DQ})}(z,t)=\frac{\partial}{\partial t}\rho_i^{(2,\mathrm{DQ})}(z,t)$. The second-order response profile is decomposed in analogy to Eq. (15) into

$$\tilde{s}_{ijk}^{(2)}(z) = \tilde{s}_{ijk}^{(2,\mathrm{D})}(z) + \tilde{s}_{ijk}^{(2,\mathrm{Q})}(z) + \tilde{s}_{ijk}^{(2,\mathrm{M})}(z)$$

(18)

$$\tilde{s}_{ijk}^{(2,\mathrm{D})}(z) = \tilde{s}_{ijk}^{(2,\mathrm{DD})}(z) + \tilde{s}_{ijk}^{(2,\mathrm{DQ})}(z),$$

(19)

where each contribution $\tilde{s}_{ijk}^{(2,\beta)}(z)$ is defined as

$$\varepsilon_0^{-1}j_i^{(2,\beta)}(z,t) = -i\omega^{\mathrm{SFG}}e^{-i\omega^{\mathrm{SFG}}t}\tilde{s}_{ijk}^{(2,\beta)}(z)\mathcal{F}_j^{\mathrm{VIS}}\mathcal{F}_k^{\mathrm{IR}} + c.c.,$$

(20)

where $\beta \in \{\mathrm{DD, DQ, D, Q, M}\}$. Consequently, the spatially integrated multipole contributions to the SFG spectrum are given by

$$\tilde{S}_{ijk}^{(2,\beta)} = \int_{-\infty}^{\infty} \mathrm{d}z\, e^{-i\Delta k_z z}\tilde{s}_{ijk}^{(2,\beta)}(z).$$

(21)

While the total current $j_i^{(2)}(z,t)$ does not depend on the choice of the molecular origin, the individual contributions in Eq. (15) do[79,115]. To determine the optimal position of the molecular origin, we consider the SFG signal from an interface with isotropically oriented molecules $\tilde{S}_{ijk}^{(2,\mathrm{ISO})}$, which we call an isotropic interface. This isotropic interface is created by cutting bulk water at an arbitrary $z$-position into two halves according to the water center of mass positions. Consequently, boundary contributions created by the change in density at the interface are present, but the molecules' orientational distribution is inversion symmetric. A similar interface was created to predict the Bethe potential[116]. We test three molecular origins for the calculation of multipoles and find that if we choose the molecular center of mass as the molecular origin, $\tilde{S}_{ijk}^{(2,\mathrm{ISO})} \approx \tilde{S}_{ijk}^{(2,\mathrm{Q})} + \tilde{S}_{ijk}^{(2,\mathrm{M})}$ does hold in good approximation. Thus, choosing the molecular center of mass as the expansion center ensures that the electric dipole contribution is solely due to molecular orientation, as discussed in detail in SI Section VI. Moreover, the center of mass maximizes the decoupling of molecular translations, vibrations, and rotations[117–119]. In contrast, choosing the molecular center different from the center of mass introduces significant boundary contributions in the electric dipole contribution $\tilde{S}_{ijk}^{(2,\mathrm{D})}$.

The theory of our multipole decomposition is described in SI Sections III and IV. There, we also derive the constitutive relations, including non-linear multipolar source terms, using the Lorentz field approximation in planar geometry, based on works by Mizrahi and

Sipe[120] and Hirano and Morita[46]. These equations are solely needed to aid the interpretation of experimental spectra.

## Fluctuation-dissipation relations within the off-resonant approximation

We assume that the VIS field interacts off-resonantly with the system, meaning it does polarize the molecules but does neither excite higher electronic levels nor influence the nuclei trajectories. This approximation is valid since the VIS field oscillates too rapidly to influence the motion of the nuclei, yet slowly enough to act adiabatically on the distribution of the electrons. In this limit, the SFG signal arises from a first-order perturbation expansion with respect to the external field from the IR light source[121].

The molecular multipoles are determined by the equations

$$\mu_i^n(t) = \alpha_{ij}^{n,\mathrm{DD}}(t)f_{jk}^n(t)F_k^{\mathrm{VIS}}(t)$$

(22)

$$Q_{ij}^n(t) = \alpha_{ijk}^{n,\mathrm{QD}}(t)f_{kl}^n(t)F_l^{\mathrm{VIS}}(t),$$

(23)

where $\mu_i^n(t)$ and $Q_{ij}^n(t)$ are the induced electric dipole and electric quadrupole moments of the $n$th-molecule, and $\alpha_{ij}^{n,\mathrm{DD}}(t)$ and $\alpha_{ijk}^{n,\mathrm{QD}}(t)$ are the electric dipole and the electric quadrupole polarizabilities of the $n$th molecule. The local field factor $f_{ij}^n(t)$ transforms an external field $F_i(t)$ into the local E-field acting on the $n$th molecule, $E_i^n(t)=f_{ij}^n(t)F_j(t)$, and is determined in a self-consistent manner such that the multipoles induced by $E_i^n(t)$ produce the field $E_i^n(t)-F_i(t)$. It is demonstrated in SI Section V B that it is incorrect to solve separate self-consistent field equations for the VIS and the SFG field in the time domain, as proposed previously[121–123]. Rather, if a timescale-separation approximation is applied, Eqs. (22) and (23) each split into components at SFG and VIS frequencies, as described in SI Section V B.

The polarization contributions of the second-order response profiles, defined in Eq. (20), are determined by a first-order perturbation expansion, for which we introduce the fluctuation-dissipation relations

$$s_{ijk}^{(2,\mathrm{D})}(z,t) = \frac{-\Theta(t)}{Ak_{\mathrm{B}}T\varepsilon_0}\frac{\partial}{\partial t}\sum_n^{N_{\mathrm{mol}}}\left\langle \alpha_{il}^{n,\mathrm{DD}}(t)f_{lj}^n(t)\delta[z-z^n(t)]P_k(0)\right\rangle$$

(24)

$$s_{ijk}^{(2,\mathrm{Q})}(z,t) = \frac{\Theta(t)}{Ak_{\mathrm{B}}T\varepsilon_0}\frac{\partial}{\partial z}\frac{\partial}{\partial t}\sum_n^{N_{\mathrm{mol}}}\left\langle \alpha_{izl}^{n,\mathrm{QD}}(t)f_{lj}^n(t)\delta[z-z^n(t)]P_k(0)\right\rangle,$$

(25)

where $\langle...\rangle$ denotes classical ensemble averaging, $T$ is the temperature, $k_B$ is the Boltzmann constant, $\Theta(x)$ is the Heaviside function, $\delta(x)$ is the Dirac delta distribution, and $z^n$ is the $z$-position of the $n$th molecule. These expressions omit the off-resonant hyperpolarizability, which can be included as shown in SI Section V, but does not contribute to the imaginary part of SFG spectra. Equations (24) and (25) follow from time-dependent perturbation theory using the perturbation Hamiltonian $H_{\mathrm{int}}(t) = -P_iF_i^{\mathrm{IR}}(t)$, where $P_i$ is the total system's dipole moment. Hirano and Morita[79] suggest the perturbation expansion using the modified perturbation Hamiltonian $H_{\mathrm{int}}^+(t) = -P_i^+ F_i^{\mathrm{IR}}(t)$, where $P_i^+$ is the dipole moment of the upper half of the simulation box. It is demonstrated in SI Section V D that this introduces a spurious contribution from the interface between the two halves.

It is shown in SI Section XII that Eqs. (24) and (25) provide a leading-order harmonic approximation of the quantum mechanical SFG response profile and that no quantum-correction factor needs to be applied. The harmonic quantum-correction factor $Q_{\mathrm{HA}} = \frac{\hbar\omega\beta}{1-e^{-\hbar\omega\beta}}$, which appears in the literature[99,124,125], relates quantum and classical time-correlation functions[124], but does not relate classical and quantum response functions.

We note that while we treat electric dipole and electric quadrupole contributions in a precise manner, we approximate magnetic dipole contributions by considering only the leading order term of an expansion in terms of electric multipole moments, as outlined in SI Section V C. We do not compute the position-resolved profile of magnetic dipole contributions, but only the overall contribution to the SFG signal, which can be extracted from a simulation of a bulk system, according to a fluctuation-dissipation relation presented in SI Section V C. Hence, we omit magnetic dipole contributions, whenever we present the response profiles $\widetilde{s}_{ijk}^{(2)}(z)$, but consider them when we present the total spectra $\widetilde{S}_{ijk}^{(2)}$. In contrast to the electric dipole and electric quadrupole contributions, the magnetic dipole contribution does depend on $\omega^{\text{VIS}}$, set to 2730 THz, which is the center frequency of the VIS-field employed in the experimental measurement of the bending band[54] we compare with. The VIS frequency in the experimental reference SFG spectra of the stretching band[34,64] is quite similar.

## Linear dielectric and absorption profiles

We define the linear response of the polarization density $p_i(z, t) = D_i(z, t) - \varepsilon_0 E_i(z, t)$ to an external field of amplitude $\mathcal{F}_i^\alpha$ as

$$\varepsilon_0^{-1} p_i^{(1)}(z, t) = e^{-i\omega^\alpha t} \widetilde{s}_{ij}^{(1, \text{P})}(z) \mathcal{F}_j^\alpha + c.c.. \tag{26}$$

The extraction of $\widetilde{s}_{ij}^{(1, \text{P})}(z)$ from molecular dynamics simulation has been described before[85-91] and is reproduced in SI Sections V B. At the interface, the dielectric profile is tensorial, and the component perpendicular to the interface is given by

$$\widetilde{\varepsilon}_{zz}^\alpha(z) = \frac{\varepsilon_0^{-1} \mathcal{D}_z^\alpha}{\mathcal{E}_z^\alpha(z)} = \frac{1}{1 - \widetilde{s}_{zz}^{(1, \text{P})}(z)}, \tag{27}$$

while the component parallel to the interface is determined by

$$\widetilde{\varepsilon}_{xx}^\alpha(z) = \frac{\varepsilon_0^{-1} \mathcal{D}_x^\alpha(z)}{\mathcal{E}_x^\alpha} = 1 + \widetilde{s}_{xx}^{(1, \text{P})}(z). \tag{28}$$

Here, $\mathcal{D}_i^\alpha(z)$ and $\mathcal{E}_i^\alpha(z)$, are defined in Eqs. (11) and (10), respectively. Due to the symmetry of our system, we have $\widetilde{\varepsilon}_{yy}^\alpha(z) = \widetilde{\varepsilon}_{xx}^\alpha(z)$. In bulk, the dielectric tensor reduces to the isotropic component

$$\widetilde{\varepsilon}^\alpha(z) = \frac{\widetilde{\varepsilon}_{xx}^\alpha(z) + \widetilde{\varepsilon}_{yy}^\alpha(z) + \widetilde{\varepsilon}_{zz}^\alpha(z)}{3}. \tag{29}$$

Further information about the dielectric profiles is given in SI Section XIII.

## Electric dipole SFG contribution as a fingerprint for interfacial structure

The response of the second-order electric current density to spatially constant external fields, which are $z$-polarized D-fields and $x$ or $y$-polarized E-fields, is given in Eq. (8). As we want to relate SFG spectra to the molecular orientation distribution, we are interested in the second-order response of the electric dipole density to the average amplitude of the local E-field $\mathcal{E}_i^{\text{L}, \alpha}(z)$ acting on the molecular centers. We define the second-order electric dipole susceptibility $\widetilde{\chi}_{ijk}^{(2, \text{DL})}(z)$ by

$$\varepsilon_0^{-1} j_i^{(2, \text{DD})}(z, t)$$
$$= -i\omega^{\text{SFG}} f_i^{\text{SFG}}(z) e^{-i\omega^{\text{SFG}} t} \widetilde{\chi}_{ijk}^{(2, \text{DL})}(z) \mathcal{E}_j^{\text{L}, \text{VIS}}(z) \mathcal{E}_k^{\text{L}, \text{IR}}(z) + c.c.. \tag{30}$$

The second-order electric dipole susceptibility $\widetilde{\chi}_{ijk}^{(2, \text{DL})}(z)$ plays a central role for the interpretation of SFG spectra as it allows to interpret the macroscopic electric dipole contribution in terms of

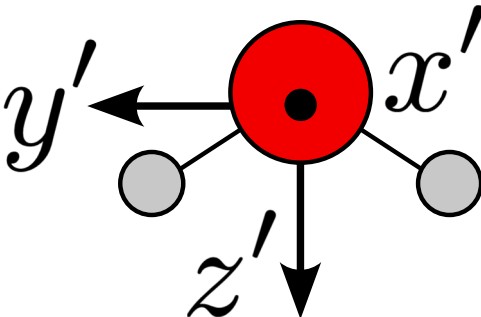

**Fig. 6 | Sketch of the molecular Eckart frame of a water molecule.** $x'$ denotes the out-of-plane axis, $z'$ the electric dipole axis, and $y'$ is orthogonal to $x'$ and $z'$.

molecular orientation[28,30-32]. $\widetilde{\chi}_{ijk}^{(2, \text{DL})}(z)$ is related to the electric dipole contribution to the SFG response profile via Eq. (1), which is a frequently employed formalism to relate macroscopic and microscopic non-linear quantities[28,45,46,73,74]. Note that we do not introduce any locality approximations in the computation of the second-order response $\widetilde{s}_{ijk}^{(2)}(z)$, but rather predict $\widetilde{\chi}_{ijk}^{(2, \text{DL})}(z)$ by dividing the nonlocal second-order response profile $\widetilde{s}_{ijk}^{(2, \text{DD})}(z)$ by the averaged local field factors $f_i^\alpha(z)$. We neglect the frequency dependence of $f_i^\alpha(z)$ by approximating $f_i^\alpha(z) \approx f_i^{\text{VIS}}(z)$ for $\alpha = \text{IR}$ and $\alpha = \text{SFG}$; this is justified because the local-field factor is dominated by its real component, which is nearly constant over the frequency range considered.

We compare $\widetilde{\chi}_{ijk}^{(2, \text{DL})}(z)$ with $\widetilde{\chi}_{ijk}^{(2, \text{ORI})}(z)$, which is solely based on the anisotropic orientation of the molecules. The Euler angles $\phi$, $\theta$, and $\psi$ specify the molecule's orientation, and we employ the $z'y'z'$ convention[126]. We define the molecular Eckart frame[117,118,127,128] by aligning the molecular $z'$-axis with the permanent dipole vector. The molecular $x'$-axis points out of the molecular plane, while the molecular $y'$-axis is chosen to be perpendicular to both $x'$ and $z'$. The molecular Eckart frame is depicted in Fig. 6. As shown in SI Section VII, the molecular hyperpolarizability $\widetilde{\beta}_{ijk}(\theta, \psi)$ is determined by three orientation-dependent functions $q_{10}(\theta)$, $q_{30}(\theta)$, and $q_{32}(\theta, \psi)$. The first and third Legendre polynomials[126]

$$q_{10}(\theta) = \cos\theta \tag{31}$$

$$q_{30}(\theta) = \frac{1}{2} \left[ 5\cos^3\theta - 3\cos\theta \right] \tag{32}$$

describe the orientation of the molecular dipoles, while $q_{32}(\theta, \psi)$ accounts for the rotation of the molecule around the molecular dipole moment and is only relevant if the molecule is not uniaxial. Consequently, $q_{32}(\theta, \psi)$ is the biaxiality parameter, given by

$$q_{32}(\theta, \psi) = \cos(\theta)(\sin^2\theta\cos^2\psi - \sin^2\theta\sin^2\psi). \tag{33}$$

We obtain $\widetilde{\chi}_{ijk}^{(2, \text{ORI})}(z)$ by averaging all hyperpolarizabilities according to

$$\widetilde{\chi}_{ijk}^{(2, \text{ORI})}(z) = \frac{1}{V} \sum_n^{N_{\text{mol}}} \widetilde{\beta}_{ijk}(\theta^n, \psi^n) \tag{34}$$

$$= \rho(z) \left[ q_{10}(z)\widetilde{\beta}_{ijk}^{10} + q_{30}(z)\widetilde{\beta}_{ijk}^{30} + q_{32}(z)\widetilde{\beta}_{ijk}^{32} \right], \tag{35}$$

where $V$ is the volume, $\rho(z)$ is the molecular density and $q_{lm}(z)$ is the $z$-dependent average of $q_{lm}(\theta^n, \psi^n)$ over all molecules. The frequency-

dependent coefficients $\widetilde{\beta}_{ijk}^{lm}$ are presented in Fig. 3j. A detailed description of the orientation analysis can be found in SI Section VII. The full information about the orientation distribution is determined by the orientation distribution function defined by

$$\rho_{\text{ORI}}(\theta, \psi) = \frac{1}{2\pi \sin \theta N_{\text{mol}}} \sum_{n}^{N_{\text{mol}}} \langle \delta(\theta - \theta^n) \delta(\psi - \psi^n) \rangle . \quad (36)$$

The prefactor ensures that $\rho_{\text{ORI}}(\theta, \psi)$ is properly normalized as

$$\int_0^{2\pi} d\phi \int_0^\pi \sin \theta d\theta \int_0^{2\pi} d\psi \, \rho_{\text{ORI}}(\theta, \psi) = 1.$$

In an isotropic system, $\rho_{\text{ORI}}(\theta, \psi)$ is constant and is given by $\frac{1}{8\pi^2}$.

## Data availability
The Source data required to reproduce the figures are provided within this paper. Data and all files necessary to reproduce the simulations have been deposited in the Zenodo repository and are available at https://doi.org/10.5281/zenodo.19055212. Source data are provided with this paper.

## Code availability
Code and a minimal dataset supporting the findings of this study have been deposited on Code Ocean and are available at https://doi.org/10.24433/CO.4603361.v1.

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

## Acknowledgements

The authors acknowledge financial support from the Deutsche Forschungsgemeinschaft (DFG) within the grant CRC 1349 (M.R.B.) and HPC resources on CURTA[129], TRON, and Sheldon from the FU Berlin for data acquisition. Discussions with F. Geiger, M. Bonn, and Y. Nagata are acknowledged.

## Author contributions

L.L. and R.R.N. conceived the theoretical framework, designed the simulations, interpreted the results, and wrote the manuscript. L.L. performed the simulations and carried out the data analysis. M.R.B. assisted with the simulation setup. L.T. implemented the Ewald summation used for calculating the local field factors. A.P.F., Á.D.D., M.T., and M.W. performed the experimental measurements. All authors discussed the results and revised the manuscript.

## Funding

## Competing interests

The authors declare no competing interests.
