## [Transparent Peer Review file · Nature Communications]

Multipolar Electric and Magnetic Contributions to Sum-Frequency Generation Spectra Reveal Biaxial Interfacial Water Structure

Corresponding Author: Professor Roland Netz

Version 0:

Reviewer comments:

Reviewer #1

(Remarks to the Author)

In the paper, the authors performed the very detailed SFG spectra calculation including the higher order contribution and addressed many questions regarding what the SFG spectra tell us. Does SFG spectra bring surface information or are SFG spectra contained by the bulk contribution? Are the SFG spectra governed by the molecular orientation at the interface (dipole contribution?). The beauty of the work is not only reproducing the lineshape of the SFG spectra but also accurately predicting the absolute amplitude of the SFG spectra. The detailed formulation starting from the fluctuation-dissipation theorem is also useful for the future works in the community. However, I found several unclear points and potentially confusing notations. I would therefore recommend the editor to publish this work after the authors address my points.

Quantum correction for classical time correlation function

If my understanding is correct, the authors did not mention the quantum correction factor, despite the fact that the O-H stretch motion ($3000-4000\text{cm}^{-1}$) is much higher than $kT=200\text{cm}^{-1}$ and therefore the O-H stretch motion is quantum. In fact, the need of considering the quantum correction for computing IR spectra has been discussed, see Eq. (2) to Eq. (3) in J.L. Skinner, et al., J. Chem. Phys. 129, 214705 (2008).

If we assume the harmonic quantum correction factor is valid ($Q \sim \hbar \omega / kT$, see Eq. (31) in D. Marx et al., J. Chem. Phys. 121, 3973 (2004)), the SFG amplitudes of the O-H stretch and O-H bend modes are enhanced by 15 times and 8 times, respectively, which is a non-negligible contribution.

Figure 3(e)

The enriched out-oriented water molecules in the topmost water layer and the enriched in-oriented water molecules in the second topmost water layer can be considered as a remaining ice Ih surface structure in the water structure. In this respect, I find the discussion is in line with the J. Am. Chem. Soc. 144, 11178 (2022).

Bulk and interfacial quadrupole contribution

The main discussion of the quadrupole contribution is whether the quadrupole contribution arises from the bulk or interface (see Ref. 46 and 48 for example for the bending mode discussion). It seems that the authors insists that the interfacial quadrupole is more than sufficient to reproduce the bend SFG feature. If so, a brief note on the discussion of Ref. 46 and 48 would be appreciated, as this is the main controversy in the bending mode discussion.

Origin of the shoulder at 3600cm^{-1}

The shoulder peak has been discussed in the following three lines. 1. Comparing H₂O and D₂O SFG data and D₂O SFG exhibits stronger shoulder peak than H₂O SFG (Tian and Shen, J. Am. Chem. Soc.). 2. Comparing H₂O and OH in D₂O SFG data and H₂O data shows the shoulder peak (Skinner and Benderskii, Nature, Tian and Shen, JACS). 3. Comparing yyz and zzz spectra, yyz spectra show a more prominent shoulder peak (Nagata et al., PNAS). The discussion on the current manuscript is closer to point 3. Now, let's compare yyz and zzz spectra. When we focus on the dipole contribution, I can say that yyz spectra show shoulder type peak than zzz. If so, the dipole contribution already shows the signature of 3600cm^{-1} peak in the yyz spectra (consistent with Skinner/Paesani/Nagata) and therefore attributing the shoulder peak all to the quadrupole contribution seems misleading.

"This is the formalism traditionally used in SFG theory [24, 25, 31, 41, 42, 67]".

When I read the paper of Shen et al., J. Chem. Phys. 161, 124117 (2024), I am afraid that a single formalism has not been accepted in the SFG theory, unlike what the authors wrote. More precisely, the fluctuation-dissipation theorem is applied by using E_{in} (input electric field) as an external field in Eq. (37) (or Eq. (5.23) in the Morita's book) and therefore the surface response is a function of IR, visible, and SFG frequencies (Eq. (38)), while in the other theory papers the surface response is not dependent on the visible and SFG frequencies, presumably because the surface response is defined in the interfacial layer and therefore the external electric field when applying the fluctuation-dissipation to the formulation is E_{loc} by using the notation of the Shen's JCP paper. Probably, the authors should be careful about the derivation based on the above different application of the fluctuation-dissipation theorem.

In addition, the current manuscript complicates the notation in my opinion. For example, Eq. (B7) is called as the "contribution to the SFG spectrum", but this is not the SFG spectrum used in many papers, but the interfacial dielectric corrected SFG spectrum. If my understanding is correct, SFG response function defined in this paper is the interfacial dielectric corrected SFG susceptibility and SFG spectra defined in this paper is the spectra of this SFG response function. (and that is why the authors' SFG spectrum cannot be directly compared with e.g. the data of Figure 5(b) in J. Chem. Phys. 144, 244711 (2016), although both are called as the SFG spectra of water-air interface). I think that it is better to name these in a way that the readers would not be confused. I was very much confused at the first glance.

Notation of "

In Figure 1, S'' is used, but in the caption, there is no S'' ; only S is used. The caption and figure should be synchronized.

(Remarks on code availability)

Reviewer #2

(Remarks to the Author)

The manuscript title "Beyond the electric dipole approximation: electric and magnetic multipole contributions reveal biaxial water structure from SFG spectra at the air-water interface" presents a theoretical method for predicting and decomposing vibrational sum-frequency generation (SFG) spectra at liquid water-air interfaces from simulation data. Previous approaches to accomplishing this task typically approximate the SFG response as originating fully from the second-order dipole susceptibility. The innovation in this work is to go beyond this approximation by including higher-order multipole contributions, including the magnetic dipole and electric quadrupole. The authors present the theory necessary to include these contributions and apply the theory to a simulation of the liquid water-air interface. The results exhibit excellent agreement with experiment and serve to highlight the important contributions that these often neglected multipole contributions make to various features of the spectra. By carefully analyzing the results, the authors derive new physical insight from existing SFG data into the depth-dependent interfacial molecular structure.

This manuscript describes a significant theoretical advance for the prediction of SFG spectra. The ability to decompose spectral into multipolar contributions will broadly enhance our interpretation of spectra and allow folks to derive more physical insight. Because the predicted spectra are derived from simulation data, the various spectral components can be spatially resolved. The authors highlight this capability with a series of interesting figures. Based on the potential for impact on the SFG community, I think this work merits publication in Nature Communications. However, there are some issues that the authors should address prior to publication. These issues are enumerated below.

1. In terms of sections and subsection, the manuscript is well organized. However, the text is very dense in technical detail and includes (in my opinion) much less clarifying exposition than is typical for a manuscript in Nature Communication. For example, there is no paragraph structuring within the manuscript (with the exception of the concluding paragraph). Because of this, it is more difficult than it ought to be to appreciate the new physical insight that is contained within the results. Some editing to highlight the physical and theoretical takeaways of each section would significantly increase the quality of the manuscript.

2. The topic of depth attenuation of the various spectral components will likely be of interest to some readers. A small section dedicated to the topic would be worth considering.

3. Minor issue: the acronym DFG is referenced in the text but never specifically defined.

(Remarks on code availability)

The code is clearly written and well commented. I was able to understand and run the code. The code includes a small sample data set derived from the larger manuscript data set. The code implies that it generates depth-resolved information. But I could not figure out how to get the code to output that information.

Reviewer #3

(Remarks to the Author)

Review of "Beyond the Electric Dipole Approximation...." By Lehmann et al.

The paper presents an analysis of the VSFS at the water-air interface. The central premise of the work is that an electric dipole alone is insufficient to explain the spectrum and that additional terms are needed. I feel the authors have overstated

this, and it is not a problem. Having examined the water spectra, I would say the community in this field does not see a theoretical problem with this, as long as the best data are used in the examination. In this paper, I do not see the authors treating the best-published spectra at high resolution, such as those by HF Wang or E. Tyrode. Their data preserves the true line shape and has been properly normalized. I would also say that the water surface is extremely susceptible to contamination, and any contamination will perturb the spectrum, which might be interpreted as effects beyond the electric dipole. Tyrode's work is considered the best, cleanest data at present. Therefore, I do not think this manuscript presents an important issue. However, the treatment is, in principle, OK.

Some details are presented below.

1. "The interpretation of sum-frequency generation (SFG) spectra has been severely limited by the absence of quantitative theoretical predictions of higher-order multipole contributions." I disagree with this
2. For the authors to be more convincing, they should compare with data beyond ssp or ppp, since these will not readily display higher-order effects. If such effects were significant, they would appear in the 'off' polarisations too, such as psp, etc., and these have not been observed.
3. The comparison in Figure 1 also highlights an important consideration. The experimental and theoretical data show a slight mismatch; however, the experimental results do not show significant error bars, especially in the phase-resolved measurements. The comparisons would then not be as significant.
4. Similarly, uncertainty values for the computation must be incorporated into the analysis.
5. The depth profiles are an interesting contribution.
6. The surest way to estimate the quadrupole and magnetic contributions is by configuration analysis and transmission experiments, which are not presented.

(Remarks on code availability)

Version 1:

Reviewer comments:

Reviewer #1

(Remarks to the Author)

The authors addressed my point fully and I would like to suggest the editor to accept the manuscript as it is.

(Remarks on code availability)

Reviewer #3

(Remarks to the Author)

Thanks to the Authors to consider my comments.

My redresses are in red below

Reviewer #3 (Remarks to the Author):

Review of "Beyond the Electric Dipole Approximation...." By Lehmann et al.

The paper presents an analysis of the VSFS at the water–air interface. The central premise of the work is that an electric dipole alone is insufficient to explain the spectrum and that additional terms are needed. I feel the authors have overstated this, and it is not a problem. Having examined the water spectra, I would say the community in this field does not see a theoretical problem with this, as long as the best data are used in the examination. In this paper, I do not see the authors treating the best-published spectra at high resolution, such as those by HF Wang or E. Tyrode. Their data preserves the true line shape and has been properly normalized. I would also say that the water surface is extremely susceptible to contamination, and any contamination will perturb the spectrum, which might be interpreted as effects beyond the electric dipole. Tyrode's work is considered the best, cleanest data at present. Therefore, I do not think this manuscript presents an important issue. However, the treatment is, in principle, OK.

Some details are presented below.

We thank the reviewer for the constructive feedback and for stressing the importance of comparing with high-quality experimental data. We understand the reviewer's concern that our conclusions could depend on the specific experimental dataset we compare with, particularly in light of possible contamination effects.

Following the reviewer's suggestion, we have added in the revised main text a comparison of our theoretical predictions with the high-resolution spectra from H. F. Wang (Zhang 25) and E. Tyrode (Sengupta 18) in Fig. 1(k). There we present the absolute SFG intensity, defined by

$\sqrt{\dots}$

Details on this comparison are presented in the newly added SI Section XI.A. The comparison in Fig. 1 (k) not only demonstrates that our theoretical multipolar framework yields quantitative agreement with the above-mentioned experimental datasets by H. F. Wang (Zhang 25) and E. Tyrode (Sengupta 18) over a broad frequency range of 1400/cm-4000/cm, it also shows that the data by H. F. Wang (Zhang 25) and E. Tyrode (Sengupta 18) is consistent with the datasets by M. Bonn (Yu 23) and M. Thämer (Fellows 25), for which we already showed the comparison of the imaginary spectra with

our predictions in the previous manuscript version in Fig. 1 (c,g).

There is some scattering among the different experimental data sets, which is expected considering the complexity of the experimental setup, and the deviation between our theoretical predictions and the experimental data is of the order of the scattering among the different experimental data sets. This comparison demonstrates the robustness of our conclusions. We are grateful to the reviewer for directing us to these datasets and for helping us to improve our manuscript.

Thank you for this addition, I think it will be helpful for all to see this comparison.

1. "The interpretation of sum-frequency generation (SFG) spectra has been severely limited by the absence of quantitative theoretical predictions of higher-order multipole contributions." I disagree with this

Let us try to explain our point of view. SFG spectroscopy is widely used as an interface-specific technique with the explicit goal of determining interfacial molecular structure. However, its ability to do so has been severely limited by the fact that it has not been known how much of the measured SFG signal arises from interfacial structure (leading to the electric dipole SFG contribution) and how much arises from bulk structure (leading to higher-order multipole SFG contributions). Due to this, interfacial structure could not be inferred from the measured SFG spectra.

To illustrate this point, we draw the reviewer's attention to Fig. 1 (e) where we present SFG data for the bending mode. There we show that the electric dipole contribution is of similar magnitude as the magnetic dipole contribution, while the electric quadrupole contribution is significantly larger than both. This vividly demonstrates that the commonly used electric-dipole approximation fails dramatically for the bending SFG signal of the air-water interface. After subtracting the multipolar contributions, Fig. 2 (c) shows that the remaining dipolar contribution enables quantitative determination of the interfacial water orientation. This is then used in Fig. 3 to extract from the SFG bending peak the biaxial interfacial water structure. Thus, our results demonstrate that the absence of quantitative theoretical predictions for higher-order multipole SFG contributions has been a serious limitation for SFG-based structural analysis and at the same time helps to overcome this limitation.

Yes, I see this part is derived from your analysis that electric dipole is not enough and that addition of other terms is needed to match the experiment. I still feel that your statement is too strong. Unfortunately there is no experimental reason to presume that the quadrupole effects are present. I note again that Tyrode and Wang both independently seem to have good modeling of their spectra based on electric dipole only.

2. For the authors to be more convincing, they should compare with data beyond ssp or ppp, since these will not readily display higher-order effects. If such effects were significant, they would appear in the 'off' polarizations too, such as psp, etc., and these have not been observed.

We thank the reviewer for this interesting suggestion. In Fig. S18 B & C in the newly added SI Section XI.B we present a comparison of our predicted and experimental SFG spectra at the air-water interface for the polarization combinations SPS and PSS using the SFG spectra from Zhang et al. [10.1063/5.0281195] and Sengupta et al. [10.1021/acs.jpcclett.8b03069]. Because SPS and PSS signals are significantly weaker than the SSP signal, which is proportional to $\chi^{(2)}$, the relative disagreement is somewhat larger than for $\chi^{(2)}$, presented in Figure 1 (k) of the manuscript. Nevertheless, our theoretical framework also works for these polarization combinations and produces better results than previous theoretical approaches (see for example Figure S4 of the SI in [10.1021/acs.jpcclett.8b03069]) that relied on empirical rescaling of the amplitude and electric dipole approximations.

3. The comparison in Figure 1 also highlights an important consideration. The experimental and theoretical data show a slight mismatch; however, the experimental results do not show significant error bars, especially in the phase-resolved measurements. The comparisons would then not be as significant.

We appreciate the reviewer's point regarding the uncertainty of the experimental data used for comparison. To address this concern, we have added Fig. 1 (k) in the main text, which compares our theoretical predictions to four independent experimental datasets. Here, we include phase-resolved (heterodyne) and non-phase-resolved experimental data. The spread among the different measurements provides a realistic estimate of the experimental uncertainty and allows the reader to assess the robustness of the comparison with our theoretical predictions, as we mention in the revised version in Section II.A.

OK

4. Similarly, uncertainty values for the computation must be incorporated into the analysis.

We thank the reviewer for this comment. We already provided 95% confidence intervals for our computed spectra in Fig. S17 of the SI in the previous version of our manuscript, which demonstrate that the signal-to-noise ratio of the simulations is high, and the resulting uncertainties are small. We have added a comment on the errors of our numerical calculation in SI Section X and a reference in the main text in Section IV.A.

OK

5. The depth profiles are an interesting contribution.

We thank the reviewer for the positive feedback and have added a brief discussion comparing the physical origins of the depth dependence of the OH-stretch and bending vibrational features at the end of Section II.D.

6. The surest way to estimate the quadrupole and magnetic contributions is by configuration analysis and transmission experiments, which are not presented.

We thank the reviewer for raising this important point. The multipole contributions addressed in our work are arising from the interface and cannot be directly extracted from experiments. Our results in fact provide the first theoretical quantification of these interfacial multipole contributions. These contributions differ fundamentally from bulk multipole contributions, which arise from external field gradients and thus depend on the experimental geometry.

As the reviewer notes, such bulk contributions can indeed be estimated experimentally through configuration analysis, reflection–transmission SFG, or combined SFG/DFG measurements. We discuss these bulk terms in SI Section IV.C. The distinction between interfacial and bulk multipole contributions was not sufficiently clear in the original manuscript, and we have improved the relevant discussion in the revised manuscript in Section IV.B.

In SSP polarization, bulk multipole terms scale with the angle between the IR and VIS wavevectors and are therefore expected to be small. However, this is not the case in PPP polarization [10.1246/bcsj.20120167].

To estimate the possible influence of bulk multipole contributions in PPP polarization, we include a comparison of our prediction with extracted experimental configuration-analysis data from Ref. [10.1063/1.2179794] in the newly added SI Fig. S18 D & E. The very good agreement between theory and experimental data demonstrates that bulk multipole contributions are not relevant and therefore do not invalidate the comparison between our predictions and experimental data shown in the main text.

Thank you for these considerations

(Remarks on code availability)

Prof. Dr. Roland Netz
Arnimallee 14
14195 Berlin

Telefon +49 30 838-55737
Fax +49 30 838-53741
E-Mail rnetz@physik.fu-berlin.de
Internet www.physik.fu-berlin.de/
einrichtungen/ag/ag-netz

10.1.2026

Manuscript "**Beyond the Electric Dipole Approximation: Electric and Magnetic Multipole Contributions Reveal Biaxial Water Structure from SFG Spectra at the Air-Water Interface**"

Dear Reviewers,

We would like to express our sincere appreciation of the constructive and insightful feedback. We made numerous changes in the main text and added further context to the main text and the Supplemental Information, that substantially strengthen our paper. Here, we reproduce the comments in full and explain the corresponding revisions made to our manuscript. Apart from minor modifications, all major additions and changes in the paper and Supplemental Information are highlighted in red in the marked-up version.

We sincerely hope that the revised version is acceptable for publication in Nature Communications.

Best wishes, the authors

Reviewer #1 (Remarks to the Author):

In the paper, the authors performed the very detailed SFG spectra calculation including the higher order contribution and addressed many questions regarding what the SFG spectra tell us. Does SFG spectra bring surface information or are SFG spectra contained by the bulk contribution? Are the SFG spectra governed by the molecular orientation at the interface (dipole contribution?). The beauty of the work is not only reproducing the lineshape of the SFG spectra but also accurately predicting the absolute amplitude of the SFG spectra. The detailed formulation starting from the fluctuation-dissipation theorem is also useful for the future works in the community. However, I found several unclear points and potentially confusing notations. I would therefore recommend the editor to publish this work after the authors address my points.

We thank the referee for the positive assessment of our work and for pointing out unclear points and notational issues. We have revised the manuscript accordingly and address each comment in detail below.

Quantum correction for classical time correlation function

If my understanding is correct, the authors did not mention the quantum correction factor, despite the fact that the O-H stretch motion (3000-4000 cm^{-1}) is much higher than $kT=200\text{cm}^{-1}$ and therefore the O-H stretch motion is quantum. In fact, the need of considering the quantum correction for computing IR spectra has been discussed, see Eq. (2) to Eq. (3) in J.L. Skinner, et al., J. Chem. Phys. 129, 214705 (2008).

If we assume the harmonic quantum correction factor is valid ($QH-\hbar\omega/kT$, see Eq. (31) in D. Marx et al., J. Chem. Phys. 121, 3973 (2004)), the SFG amplitudes of the O-H stretch and O-H bend modes are enhanced by 15 times and 8 times, respectively, which is a non-negligible contribution.

We thank the reviewer for raising this important point. Indeed, since $\hbar\omega \gg kT$, stretch and bend vibrations are essentially quantum and consequently the harmonic quantum correction factor is significantly larger than one. Therefore, at first sight, it seems unclear why we obtain such a good agreement between experimental spectra and our theoretical predictions, which are based on classical response theory. We added a short explanation in the main text in Methods Section IV.D and the entirely new Section XII.B in the SI, where we show that to leading order quantum and classical SFG spectra coincide. The reasoning is as follows:

The SFG spectrum is a response function of the polarizability to an external electromagnetic field and is calculated from (classical or quantum) correlation functions using the (classical or quantum) fluctuation-dissipation theorem (FDT) [R. Kubo, Reports on Progress in Physics 29, 255–284 (1966)]. It is important that one uses a consistent theoretical framework, so either one has to use classical correlation functions in conjunction with the classical FDT or quantum correlation functions in conjunction with the quantum FDT. In fact, to leading order in a non-harmonic expansion of the bond potential the response functions calculated via the classical and quantum routes and consequently the SFG spectra are identical. Indeed, in the harmonic approximation quantum correction factors can be used to relate classical and quantum correlation functions, as described in [D. Marx et al., J. Chem. Phys. 121, 3973 (2004)], but, as explained above, in a consistent theoretical framework the resultant spectra will not change. In fact, Equation (3) in the reference [J.L. Skinner, et al., J. Chem. Phys. 129, 214705 (2008)] results from inserting a quantum-corrected correlation function into the classical FDT, which is inconsistent.

Figure 3(e)

The enriched out-oriented water molecules in the topmost water layer and the enriched in-oriented water molecules in the second topmost water layer can be considered as a remaining ice Ih surface structure in the water structure. In this respect, I find the discussion is in line with the J. Am. Chem. Soc. 144, 11178 (2022).

This is an excellent point. In Figure 3, we present the biaxial triple-layer interfacial water structure together with its spectroscopic signatures, which indeed is similar to the structure found at the ice–air interface, as discussed in J. Am. Chem. Soc. 144, 11178 (2022). We now mention this nice connection in the revised manuscript in the Conclusion Section III and cite the suggested paper.

Bulk and interfacial quadrupole contribution

The main discussion of the quadrupole contribution is whether the quadrupole contribution arises from the bulk or interface (see Ref. 46 and 48 for example for the bending mode discussion). It seems that the authors insist that the interfacial quadrupole is more than sufficient to reproduce the bend SFG feature. If so, a brief note on the discussion of Ref. 46 and 48 would be appreciated, as this is the main controversy in the bending mode discussion.

We thank the reviewer for raising this important point. We agree that it is helpful for the readership to briefly recapitulate the literature discussion regarding Refs. 46 [Kundu et al.] and 48 [Ahmed et al.] in our paper. The quadrupole contributions considered by Ahmed et al. are discussed in SI Section IV.B and are eliminated by our nonlocal formalism, according to which the SFG profile $\tilde{s}_{ijk}^{(2)}(z)$ is written in terms of spatially constant z-polarized D-fields and x/y-polarized E-fields, both of which are gradient-free. Our results are therefore consistent with the normal-mode calculations from Kundu et al., which predict a dominant role of the quadrupole contribution, as we explicitly note in the revised version of the main text in Section II.A.

Origin of the shoulder at 3600 cm^{-1}

The shoulder peak has been discussed in the following three lines. 1. Comparing H₂O and D₂O SFG data and D₂O SFG exhibits stronger shoulder peak than H₂O SFG (Tian and Shen, J. Am. Chem. Soc.). 2.

Comparing H₂O and OH in D₂O SFG data and H₂O data shows the shoulder peak (Skinner and Benderskii, Nature, Tian and Shen, JACS). 3. Comparing yyz and zzz spectra, yyz spectra show a more prominent shoulder peak (Nagata et al., PNAS). The discussion on the current manuscript is closer to point 3. Now, let's compare yyz and zzz spectra. When we focus on the dipole contribution, I can say that yyz spectra show shoulder type peak than zzz. If so, the dipole contribution already shows the signature of 3600cm⁻¹ peak in the yyz spectra (consistent with Skinner/Paesani/Nagata) and therefore attributing the shoulder peak all to the quadrupole contribution seems misleading.

We thank the reviewer for the detailed discussion of previous work. The isotopic dilution studies mentioned by the reviewer [10.1021/ja809497y, 10.1038/nature10173] indeed report a pronounced change of the shoulder with isotopic dilution. Since we did not apply our formalism to isotope effects yet, we can at this point not contribute to this discussion.

Our results are, however, consistent with the observation that the shoulder feature is more pronounced in yyz than in zzz total SFG spectra [10.1073/pnas.2204156119, 10.1063/5.0133428]. This can be seen in Fig. 1 (g) & (h), where our predictions agree well with experiment. We also fully agree that the yyz element of the dipole contribution at 3600 cm⁻¹ shows a small shoulder, while the zzz element does not. At the same time, our analysis indicates that the amplitude of the positive shoulder in the total SFG spectrum near 3600 cm⁻¹ is mainly associated with the quadrupole term for both yyz and zzz. In this sense, the distinction between “causing” and “enhancing” the shoulder is partly a matter of perspective, and we have softened the wording in the abstract and added a clarifying sentence in Section II.A to avoid any potential misunderstanding.

“This is the formalism traditionally used in SFG theory [24, 25, 31, 41, 42, 67]”.

When I read the paper of Shen et al., J. Chem. Phys. 161, 124117 (2024), I am afraid that a single formalism has not been accepted in the SFG theory, unlike what the authors wrote. More precisely, the fluctuation-dissipation theorem is applied by using E_{in} (input electric field) as an external field in Eq. (37) (or Eq. (5.23) in the Morita's book) and therefore the surface response is a function of IR, visible, and SFG frequencies (Eq. (38)), while in the other theory papers the surface response is not dependent on the visible and SFG frequencies, presumably because the surface response is defined in the interfacial layer and therefore the external electric field when applying the fluctuation-dissipation to the formulation is E_{loc} by using the notation of the Shen's JCP paper. Probably, the authors should be careful about the derivation based on the above different application of the fluctuation-dissipation theorem.

In addition, the current manuscript complicates the notation in my opinion. For example, Eq. (B7) is called as the “contribution to the SFG spectrum”, but this is not the SFG spectrum used in many papers, but the interfacial dielectric corrected SFG spectrum. If my understanding is correct, SFG response function defined in this paper is the interfacial dielectric corrected SFG susceptibility and SFG spectra defined in this paper is the spectra of this SFG response function. (and that is why the authors' SFG spectrum cannot be directly compared with e.g. the data of Figure 5(b) in J. Chem. Phys. 144, 244711 (2016), although both are called as the SFG spectra of water-air interface). I think that it is better to name these in a way that the readers would not be confused. I was very much confused at the first glance.

We thank the reviewer for pointing this out. We agree that different theoretical approaches exist in the SFG literature, particularly regarding how the fluctuation–dissipation theorem is applied. To add to the confusion, different definitions of $\tilde{\chi}_{ijk}^{(2)}(z)$ coexist in literature. In the literature mainly two different meanings of $\tilde{\chi}_{ijk}^{(2)}(z)$ can be found:

- i) Macroscopic Susceptibility $\tilde{\chi}_{ijk}^{(2)}(z)$: The second-order susceptibility, where D-fields are translated to E-fields by the effective interfacial dielectric constant
- ii) Microscopic Susceptibility $\tilde{\chi}_{ijk}^{(2,DL)}(z)$: The density of molecular hyperpolarizabilities and a response function to local fields that act on the molecular centers. In our work we employ this formalism in the analysis of the orientation distribution, but not for SFG spectra prediction. We name this function $\tilde{\chi}_{ijk}^{(2,DL)}(z)$ (dipole, local) to distinguish it from the macroscopic susceptibility (i).

In our nonlocal formalism, the SFG response profile $\tilde{s}_{ijk}^{(2)}(z)$ corresponds to the response to z-polarized D-fields and x/y-polarized E-fields, both of which are uniform on molecular length scales. Therefore, $\tilde{s}_{ijk}^{(2)}(z)$ is not a susceptibility in the standard electrodynamic sense, which is defined as a response to an E-field; for this reason, we compare our calculated SFG spectra with the ϵ' -uncorrected experimental SFG spectra, which we took from [10.1073/pnas.2204156119] and [10.1063/5.0133428]. We now clarify the relationship between $\tilde{s}_{ijk}^{(2)}(z)$ and $\tilde{\chi}_{ijk}^{(2)}(z)$ explicitly in Equations (12)-(13) in the revised manuscript.

Finally, to avoid the impression that a single formalism is universally adopted, we have revised the phrasing in our manuscript. Instead of referring to our formalism as “the formalism traditionally used in SFG theory,” we now describe our formalism as “frequently employed formalism to relate macroscopic and microscopic nonlinear quantities” in Section IV.F.

Notation of “

In Figure 1, S” is used, but in the caption, there is no S”; only S is used. The caption and figure should be synchronized.

We thank the reviewer for pointing this out. We have corrected the inconsistency between the notation in Figure 1 and its caption so that both now use the same symbol. The same correction has been applied to the other figures.

#####

Reviewer #2 (Remarks to the Author):

The manuscript title “Beyond the electric dipole approximation: electric and magnetic multipole contributions reveal biaxial water structure from SFG spectra at the air-water interface” presents a theoretical method for predicting and decomposing vibrational sum-frequency generation (SFG) spectra at liquid water-air interfaces from simulation data. Previous approaches to accomplishing this task typically approximate the SFG response as originating fully from the second-order dipole susceptibility. The innovation in this work is to go beyond this approximation by including higher-order multipole contributions, including the magnetic dipole and electric quadrupole. The authors present the theory necessary to include these contributions and apply the theory to a simulation of the liquid water-air interface. The results exhibit excellent agreement with experiment and serve to highlight the important contributions that these often neglected multipole contributions make to various features of the spectra. By carefully analyzing the results, the authors derive new physical insight from existing SFG data into the depth-dependent interfacial molecular structure.

This manuscript describes a significant theoretical advance for the prediction of SFG spectra. The ability to decompose spectral into multipolar contributions will broadly enhance our interpretation of spectra and allow folks to derive more physical insight. Because the predicted spectra are derived from simulation data, the various spectral components can be spatially resolved. The authors highlight this capability with a series of interesting figures. Based on the potential for impact on the SFG community, I think this work merits publication in Nature Communications. However, there are some issues that the authors should address prior to publication. These issues are enumerated below.

We thank the referee for the summary of our work and for the detailed comments, which are answered below.

1. In terms of sections and subsection, the manuscript is well organized. However, the text is very dense in technical detail and includes (in my opinion) much less clarifying exposition than is typical for a manuscript in Nature Communication. For example, there is no paragraph structuring within the manuscript (with the exception of the concluding paragraph). Because of this, it is more difficult than it ought to be to appreciate the new physical insight that is contained within the results. Some editing to highlight the physical and theoretical takeaways of each section would significantly increase the quality of the manuscript.

We added short summarizing comments at the end of Sections II.A, II.C and II.D to clarify the physical insights and to present the key advances in accessible terms, thereby enhancing the manuscript’s accessibility to a wider readership. Further, at the end of Section I we provide a brief overview of the content of our manuscript.

2. The topic of depth attenuation of the various spectral components will likely be of interest to some readers. A small section dedicated to the topic would be worth considering.

We agree that the depth attenuation of the different spectral components is of interest. We now explicitly compare the depth attenuation of the stretching and bending features and discuss the physical origins of these differences at the end of Section II.D.

3. Minor issue: the acronym DFG is referenced in the text but never specifically defined.

We thank the referee for pointing this out, and we have now defined the acronym DFG in the manuscript.

#####

Reviewer #2 (Remarks on code availability):

The code is clearly written and well commented. I was able to understand and run the code. The code includes a small sample data set derived from the larger manuscript data set. The code implies that it generates depth-resolved information. But I could not figure out how to get the code to output that information.

We thank the reviewer for the positive feedback on our code. In the revised code, we include a demonstration that outputs a spatially resolved SFG-profile.

#####

Reviewer #3 (Remarks to the Author):

Review of "Beyond the Electric Dipole Approximation..." By Lehmann et al.

The paper presents an analysis of the VSFS at the water-air interface. The central premise of the work is that an electric dipole alone is insufficient to explain the spectrum and that additional terms are needed. I feel the authors have overstated this, and it is not a problem. Having examined the water spectra, I would say the community in this field does not see a theoretical problem with this, as long as the best data are used in the examination. In this paper, I do not see the authors treating the best-published spectra at high resolution, such as those by HF Wang or E. Tyrode. Their data preserves the true line shape and has been properly normalized. I would also say that the water surface is extremely susceptible to contamination, and any contamination will perturb the spectrum, which might be interpreted as effects beyond the electric dipole. Tyrode's work is considered the best, cleanest data at present. Therefore, I do not think this manuscript presents an important issue. However, the treatment is, in principle, OK.

Some details are presented below.

We thank the reviewer for the constructive feedback and for stressing the importance of comparing with high-quality experimental data. We understand the reviewer's concern that our conclusions could depend on the specific experimental dataset we compare with, particularly in light of possible contamination effects.

Following the reviewer's suggestion, we have added in the revised main text a comparison of our theoretical predictions with the high-resolution spectra from H. F. Wang (Zhang 25) and E. Tyrode (Sengupta 18) in Fig. 1(k)). There we present the absolute SFG intensity, defined by

$$|\tilde{S}_{ijk}^{(2)}| = \sqrt{\left(\tilde{S}_{ijk}^{(2)'}\right)^2 + \left(\tilde{S}_{ijk}^{(2)''}\right)^2}.$$

Details on this comparison are presented in the newly added SI Section XI.A. The comparison in Fig. 1 (k) not only demonstrates that our theoretical multipolar framework yields quantitative agreement with the above-mentioned experimental datasets by H. F. Wang (Zhang 25) and E. Tyrode (Sengupta 18) over a broad frequency range of 1400/cm-4000/cm, it also shows that the data by H. F. Wang (Zhang 25) and E. Tyrode (Sengupta 18) is consistent with the datasets by M. Bonn (Yu 23) and M. Thämer (Fellows 25), for which we already showed the comparison of the imaginary spectra with our predictions in the previous manuscript version in Fig. 1 (c,g). There is some scattering among the different experimental data sets, which is expected

considering the complexity of the experimental setup, and the deviation between our theoretical predictions and the experimental data is of the order of the scattering among the different experimental data sets. This comparison demonstrates the robustness of our conclusions. We are grateful to the reviewer for directing us to these datasets and for helping us to improve our manuscript.

1. "The interpretation of sum-frequency generation (SFG) spectra has been severely limited by the absence of quantitative theoretical predictions of higher-order multipole contributions." I disagree with this

Let us try to explain our point of view. SFG spectroscopy is widely used as an interface-specific technique with the explicit goal of determining interfacial molecular structure. However, its ability to do so has been severely limited by the fact that it has not been known how much of the measured SFG signal arises from interfacial structure (leading to the electric dipole SFG contribution) and how much arises from bulk structure (leading to higher-order multipole SFG contributions). Due to this, interfacial structure could not be inferred from the measured SFG spectra.

To illustrate this point, we draw the reviewer's attention to Fig. 1 (e) where we present SFG data for the bending mode. There we show that the electric dipole contribution is of similar magnitude as the magnetic dipole contribution, while the electric quadrupole contribution is significantly larger than both. This vividly demonstrates that the commonly used electric-dipole approximation fails dramatically for the bending SFG signal of the air–water interface. After subtracting the multipolar contributions, Fig. 2 (c) shows that the remaining dipolar contribution enables quantitative determination of the interfacial water orientation. This is then used in Fig. 3 to extract from the SFG bending peak the biaxial interfacial water structure. Thus, our results demonstrate that the absence of quantitative theoretical predictions for higher-order multipole SFG contributions has been a serious limitation for SFG-based structural analysis and at the same time helps to overcome this limitation.

2. For the authors to be more convincing, they should compare with data beyond ssp or ppp, since these will not readily display higher-order effects. If such effects were significant, they would appear in the 'off' polarisations too, such as psp, etc., and these have not been observed.

We thank the reviewer for this interesting suggestion. In Fig. S18 B & C in the newly added SI Section XI.B we present a comparison of our predicted and experimental SFG spectra at the air-water interface for the polarization combinations SPS and PSS using the SFG spectra from Zhang et al. [10.1063/5.0281195] and Sengupta et al. [10.1021/acs.jpcllett.8b03069]. Because SPS and PSS signals are significantly weaker than the SSP signal, which is proportional to $|\xi_{yyz}^{(2)}|$, the relative disagreement is somewhat larger than for $|\xi_{yyz}^{(2)}|$, presented in Figure 1 (k) of the manuscript. Nevertheless, our theoretical framework also works for these polarization combinations and produces better results than previous theoretical approaches (see for example Figure S4 of the SI in [10.1021/acs.jpcllett.8b03069]) that relied on empirical rescaling of the amplitude and electric dipole approximations.

3. The comparison in Figure 1 also highlights an important consideration. The experimental and theoretical data show a slight mismatch; however, the experimental results do not show significant error bars, especially in the phase-resolved measurements. The comparisons would then not be as significant.

We appreciate the reviewer's point regarding the uncertainty of the experimental data used for comparison. To address this concern, we have added Fig. 1 (k) in the main text, which compares our theoretical predictions to four independent experimental datasets. Here, we include phase-resolved (heterodyne) and non-phase-resolved experimental data. The spread among the different measurements provides a realistic estimate of the experimental uncertainty

and allows the reader to assess the robustness of the comparison with our theoretical predictions, as we mention in the revised version in Section II.A.

4. Similarly, uncertainty values for the computation must be incorporated into the analysis.

We thank the reviewer for this comment. We already provided 95% confidence intervals for our computed spectra in Fig. S17 of the SI in the previous version of our manuscript, which demonstrate that the signal-to-noise ratio of the simulations is high, and the resulting uncertainties are small. We have added a comment on the errors of our numerical calculation in SI Section X and a reference in the main text in Section IV.A.

5. The depth profiles are an interesting contribution.

We thank the reviewer for the positive feedback and have added a brief discussion comparing the physical origins of the depth dependence of the OH-stretch and bending vibrational features at the end of Section II.D.

6. The surest way to estimate the quadrupole and magnetic contributions is by configuration analysis and transmission experiments, which are not presented.

We thank the reviewer for raising this important point. The multipole contributions addressed in our work are arising from the interface and cannot be directly extracted from experiments. Our results in fact provide the first theoretical quantification of these interfacial multipole contributions. These contributions differ fundamentally from bulk multipole contributions, which arise from external field gradients and thus depend on the experimental geometry.

As the reviewer notes, such bulk contributions can indeed be estimated experimentally through configuration analysis, reflection–transmission SFG, or combined SFG/DFG measurements. We discuss these bulk terms in SI Section IV.C. The distinction between interfacial and bulk multipole contributions was not sufficiently clear in the original manuscript, and we have improved the relevant discussion in the revised manuscript in Section IV.B.

In SSP polarization, bulk multipole terms scale with the angle between the IR and VIS wavevectors and are therefore expected to be small. However, this is not the case in PPP polarization [10.1246/bcsj.20120167].

To estimate the possible influence of bulk multipole contributions in PPP polarization, we include a comparison of our prediction with extracted experimental configuration-analysis data from Ref. [10.1063/1.2179794] in the newly added SI Fig. S18 D & E. The very good agreement between theory and experimental data demonstrates that bulk multipole contributions are not relevant and therefore do not invalidate the comparison between our predictions and experimental data shown in the main text.

We thank the reviewer for the helpful comment. Please find below our response to the final comment.

Kind regards,

The authors

[Comment 2]

Yes, I see this part is derived from your analysis that electric dipole is not enough and that addition of other terms is needed to match the experiment. I still feel that your statement is too strong. Unfortunately there is no experimental reason to presume that the quadrupole effects is present. I note again that Tyrode and Wang both independently seem to have good modeling of their spectra based on electric dipole only.

[Reply 2]

Indeed, Tyrode observes in the publication [10.1021/acs.jpcclett.8b03069] in Figure 4 good agreement between simulation and experiment in the OH-stretch region but shows in the same figure that simulations completely disagree with experimental data in the bending region. We do not claim that all literature simulation SFG spectra are wrong, but rather that for some vibration bands the quadrupolar contribution, which has been neglected in most previously published simulated spectra, is dominant. We removed the phrase "severely limited" from the revised abstract and rather focus on the positive features of our work.